# A double-agent microRNA regulates viral cross-kingdom infection in animals and plants

Wan Zhao [1,2✉], Hong, Lu [1], Jiaming Zhu [1], Lan Luo[1] & Feng Cui [1,2✉]

## Abstract

**Plant arbovirus infection is regulated by a delicate interplay between virus, vector, and host. While microRNAs are known to be transmitted across species, their role as cross-kingdom effectors in influencing arbovirus infectious cycles remains poorly understood. Our study reveals the dual role of miR-263a, a conserved insect microRNA, in governing rice stripe virus (RSV) infection within both insect vector, small brown planthopper, and rice host. In the planthopper, miR-263a facilitates rice stripe virus accumulation through targeting a *cathepsin B-like* gene to inhibit apoptosis in midgut epithelial cells. Upon insect saliva secretion, miR-263a is delivered into rice, where it proceeds to upregulate the transcription factor *GATA19*, triggering an antiviral response. The increase of GATA19 levels hinders JAZ1 from binding with MYC2, thus activating jasmonate signaling pathway. This study reveals the function of a microRNA as a dual agent in modulating viral cross-kingdom infection.**

**Keywords** MicroRNA; Plant Arbovirus; Insect Vector; Apoptosis; Jasmonate Signal Pathway
**Subject Categories** Microbiology, Virology & Host Pathogen Interaction; Plant Biology; RNA Biology

## Introduction

Compared with the viruses independent on vectors for transmission, plant and animal arboviruses have a more complicate infection cycle, which is involved in virus-vector-host tripartite interaction and key for outbreaks of arboviral diseases. Insects occupy the majority in arbovirus vectors. It is generally believed that arboviruses induce severe symptoms primarily in hosts rather than insect vectors (Ma et al, 2021; Zhao et al, 2016b), with the virus and insect often collaborating for successful infection (Ingwell et al, 2012). Keeping a limited titer or replication level is vital for arboviruses to maintain a friendly relationship with insect vector, while viral amount in host always attains to excessiveness at certain infection period. The surveillance of host/vector immune systems and their interaction with viruses play an important role in controlling viral titers. MicroRNAs (miRNAs) are 19 to 24 nt of small non-coding RNAs and belong to RNA interference (RNAi) pathway, which is one of the most essential antiviral immune systems existing in both plants and animals, especially insects (Kanakala and Ghanim, 2016; Nayak et al, 2013). Although miRNAs are well known in regulating viral infection in either host or insect vector (Hussain et al, 2012; Singh et al, 2022; Yan et al, 2014; Zheng et al, 2013), how they affect the virus-vector-host tripartite relation as cross-kingdom effectors is almost elusive.

Since the discovery that miRNAs remain stable even in extracellular environments such as serum and blood in 2008 (Chen et al, 2008), instances of miRNA transfer across species have become increasingly documented. Plant miRNAs were found to modulate gene expression in human and mice upon consumption, whether via regular food intake or Chinese medicinal preparations (Dickinson et al, 2013; Zhang et al, 2012). Similarly, plant miRNAs are also transferred into insects; deep sequencing uncovered 13 sorghum miRNAs and three barley miRNAs in both green bugs and yellow sugarcane aphids (Wang et al, 2017). In addition, more than 200 *Brassica oleracea* miRNAs were identified in the gut sRNA library of green peach aphids (Thompson et al, 2019). On the contrary, cases of insect miRNA transmission to plants during feeding are scarcely reported. Recently, an insect-derived miRNA, miR-7-5P, of rice planthopper *Nilaparvata lugens* was found to be secreted into host plants and suppress plant resistance against planthoppers via targeting host genes (Zhang et al, 2024). Numerous miRNAs have been detected in the saliva of human arboviral vectors through small RNA sequencing, such as 320 miRNAs in the tick *Haemaphysalis longicornis* (Malik et al, 2019) and 77 miRNAs in the mosquito *Anopheles coluzzii* (Arca et al, 2019). Yet, it remains unknown whether these insect-derived miRNAs can serve as cross-kingdom effectors to regulate arboviral infection in hosts.

More than 80% of plant viruses depend on insect vectors for their transmission, causing agricultural losses of up to 60 billion annually, significantly threatening global food security (Fereres and Moreno, 2009). Among them, rice stripe virus (RSV) stands as one of the most notorious pathogens to rice crops. Classified under the *Tenuivirus* genus, RSV has a complex genome consisting of four RNA segments, encoding a total of seven proteins: these include a nucleocapsid protein (NP), an RNA-dependent RNA polymerase (RdRp), and five nonstructural proteins (Zhao et al, 2019b). RSV is efficiently transmitted by the small brown planthopper (SBPH), *Laodelphax striatellus*, in a persistent, propagative manner (Toriyama, 1986).

[1]State Key Laboratory of Integrated Management of Pest Insects and Rodents, Institute of Zoology, Chinese Academy of Sciences, Beijing 100101, China. [2]University of Chinese Academy of Sciences, Beijing 100049, China. ✉E-mail: zhaow@ioz.ac.cn; cuif@ioz.ac.cn

Remarkably, while RSV is capable of replicating in both SBPHs and rice plants, its pathogenic effects are confined to rice hosts, a testament to the distinct adaptive mechanisms employed in these two organisms (Zhao et al, 2019b; Zhao et al, 2018; Zhao et al, 2022a). Our previous work revealed that insect miR-263a promotes RSV replication in SBPHs through binding to the extended 3′ terminal region of viral genomic RNA1 segment (Zhao et al, 2021). However, miR-263a is significantly downregulated by RSV through the dual action of RSV NP and RSV-derived small RNAs (Zhao et al, 2022a). Actually miR-263a has been extensively detected in arthropods, especially in insects, such as *Aedes aegypti* (Rodríguez-Sanchez et al, 2021; Saldana et al, 2017), *Manduca sexta* (Zhang et al, 2014), and *Acyrthosiphon pisum* (Legeai et al, 2010), but not in plants or mammals. Furthermore, miR-263a was even detected in the saliva of arthropod vectors, including *Anopheles coluzzii*, *Ae. aegypti*, *Aedes albopictus*, and *H. longicornis* (Arca et al, 2019). Given its ubiquitous presence in insects and secretion via saliva, we hypothesize that miR-263a would have fundamental effects on the virus-vector-host tripartite relation by regulating the expression of insect endogenous genes and host genes.

To clarify this hypothesis, in this work, we explored the functions of miR-263a in plant arbovirus transmission with the RSV-SBPH-rice system. Our findings revealed that miR-263a exerts dual functionality: it not only governs the apoptotic activity within the insect cells but is also exported into rice cells, where it influences phytohormone balance, producing profound impacts on the interactions of virus, vector and host.

# Results

## MiR-263a benefits RSV accumulation in SBPH

The mature miR-263a is 24 nt, with a sequence of 5′-AAUGGCA-CUGGAAGAAUUCACGGG-3′. Northern blot assay confirmed the presence of primary, precursor, and mature miR-263a exclusively in SBPHs, not in rice (Fig. 1A). A comparative study of miR-263a expression between viruliferous and nonviruliferous SBPHs revealed a reduction of miR-263a in the guts and salivary glands of viruliferous insects, while the decline was insignificant across the whole bodies of 5 insects using real-time quantitative PCR (qPCR) (Fig. 1B). We applied a northern blot assay to compare miR-263a levels in the whole bodies of 30 viruliferous and nonviruliferous SBPHs. Increasing the sample size revealed a significant decrease in miR-263a expression in the bodies of the viruliferous SBPHs (Fig. 1C). To further investigate miR-263a's function, chemically synthesized miR-263a agomir (to enhance expression) and antagomir (to inhibit expression) were introduced separately into nonviruliferous SBPH along with RSV crude extracts. qPCR showed that the amount of RSV in terms of viral *NP* RNA level increased dramatically accompanying with the upregulation of miR-263a and dropped down with the reduction of miR-263a at 4 d post injection of miR-263a agomir or antagomir (Fig. 1D,E). These results indicate that miR-263a benefits RSV accumulation in SBPH but it is downregulated in insect guts and salivary glands during RSV infection.

## MiR-263a targets *Cathepsin B-like* to favor RSV accumulation in SBPH

To investigate the mechanism by which miR-263a benefits RSV accumulation in SBPH, the candidate target genes of miR-263a

were predicted in the full-length transcriptome of SBPH using miRanda and RNAhybrid algorithms (Tong et al, 2022). Totally 38 genes were predicted as candidate targets (Table EV1), among which five genes displayed differential expression following RSV infection in a gut transcriptome study: *Cathepsin B-like* (*CathBL*; PB.8570.2 in the full-length transcriptome), *serine protease snake-5* (*SPS5*; PB.9743.1), *uncharacterized protein Dsimw501* (*Dsimw501*; PB.8937.1), *UDP-glucuronosyltransferase 2B5* (*UGT2B5*; PB.5549.1) and *nuclear transcription factor Y subunit beta-like* (*NTFYbL*; PB.4359.9) (Table EV1) (Zhao et al, 2016a). Given the gut-specific response of miR-263a to RSV infection, these five differentially expressed genes in the gut were prioritized for further validation as miR-263a targets.

In nonviruliferous SBPH injected with miR-263a agomir, the transcript levels of *CathBL* and *SPS5* were reduced, whereas *NTFYbL* expression increased at 4 d post injection (Fig. 2A). Conversely, injection of miR-263a antagomir reverted these expression patterns of the three genes (Fig. 2B). The remaining two candidates did not exhibit changes in response to miR-263a modulation. Computational predictions placed miR-263a binding sites within the 5′UTRs of *CathBL* and *SPS5*, and the 3′UTR of *NTFYbL* (Fig. EV1). Direct interaction of miR-263a with these target sites was validated in *Drosophila* S2 cells by dual luciferase assays. Plasmids containing *CathBL* 5′UTR target site displayed reduced luciferase activity when cotransfected with 10 nM or 50 nM miR-263a mimic, relative to controls (Fig. EV2). Conversely, the miR-263a mimic failed to alter luciferase output from plasmids containing *SPS5* or *NTFYbL* target sequences (Fig. EV2). Introducing mutations in the *CathBL* target site that disrupted base pairing with the miR-263a seed region abolished miR-263a's suppressive effect (Fig. 2C), affirming the direct binding of miR-263a with *CathBL* 5′UTR. An RNA immunoprecipitation followed by qPCR (RIP-qPCR) experiment, employing an anti-Ago1 monoclonal antibody in nonviruliferous SBPH, revealed a significant enrichment of the *CathBL* 5′UTR upon miR-263a agomir injection compared to the control agomir injection (Fig. 2D). To clarify whether the binding of miR-263a to the 5′UTR of *CathBL* induces the degradation of *CathBL* mRNA, we performed a northern blot assay in S2 cells, in which *CathBL* mRNA containing the 5′UTR was expressed with co-transfection of miR-263a mimic or NC. A significant reduction in full-length *CathBL* mRNA and an increase in degraded fragments were observed when the miR-263a mimic was present, as detected with a biotin-labeled probe specific for *CathBL* (Fig. 2E). Collectively, these data confirm that miR-263a downregulates *CathBL* expression in SBPH via directly interacting with its 5′UTR.

To explore the effect of *CathBL* on RSV accumulation, a 226-bp double-strand RNA (dsRNA) specific to *CathBL* was introduced into viruliferous fourth-instar SBPH larvae. After 4 d of injection with ds*CathBL*-RNA, which effectively suppressed *CathBL* expression, the RNA and protein level of RSV NP significantly increased in SBPH compared to the injection of ds*GFP*-RNA (Fig. 2F,G), indicating that *CathBL* plays a negative role in RSV accumulation in insects.

## Cathepsin B-like induces a caspase-dependent apoptosis in midgut epithelial cells

The open reading frame (ORF) of SBPH *CathBL* spans 1041 bp, encoding a predicted 39-kDa protein (GenBank number

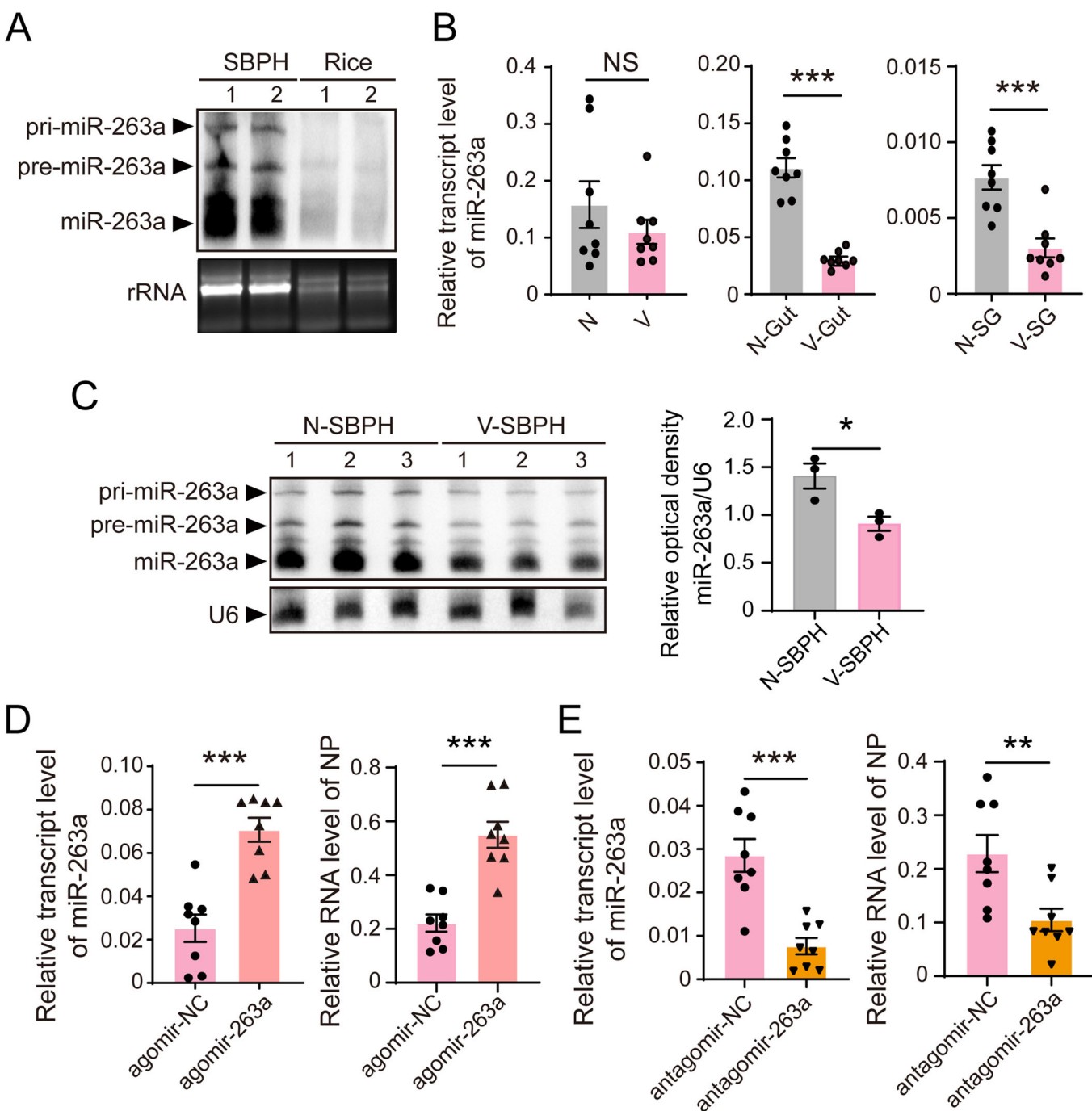

**Figure 1. MiR-263a benefits RSV accumulation in SBPH.**

(A) Northern blot analysis for determining the existence of miR-263a in nonviruliferous SBPH and healthy rice leaves; the signal was detected using a biotin-labeled LNA oligonucleotide probe for miR-263a. rRNA from SBPHs and rice was used as an internal control. Each sample includes two replicates. (B) Relative transcript level of miR-263a in the whole bodies, guts, and salivary glands (SGs) of nonviruliferous (N) and viruliferous (V) SBPH quantified by qPCR. SBPH U6 snRNA was used as an endogenous control. Eight biological replicates were prepared, with each replicate consisting of 5 insect bodies, 8 guts, or 10 salivary glands. $P$ values from left to right, $P = 1.48E-05$, $P = 4.00E-04$. (C) Northern blot showing the transcript levels of miR-263a in the whole bodies of nonviruliferous and viruliferous SBPHs. SBPH U6 snRNA was used as an internal control. Each sample includes three replicates. $P = 0.0296$. (D, E) Relative transcript level of miR-263a and the RNA level of viral *NP* in nonviruliferous fourth-instar SBPH larvae after injected with the mixture of RSV crude extractions and agomir-263a/agomir-NC (D), or the mixture of RSV crude extractions and antagomir-263a/antagomir-NC (E) for 4 d. SBPH U6 snRNA and *EF2* were used as an endogenous control for miR-263a and viral *NP*, respectively. Eight biological replicates were prepared, with each replicate consisting of 5 insect bodies. $P$ values from left to right, $P = 9.67E-05$, $P = 5.80E-05$, $P = 2.00E-04$, $P = 0.0085$. For (B) to (E), values are presented as mean ± SE and were compared using Student's t test. NS, no significant difference. *$P < 0.05$. **$P < 0.01$. ***$P < 0.001$. Source data are available online for this figure.

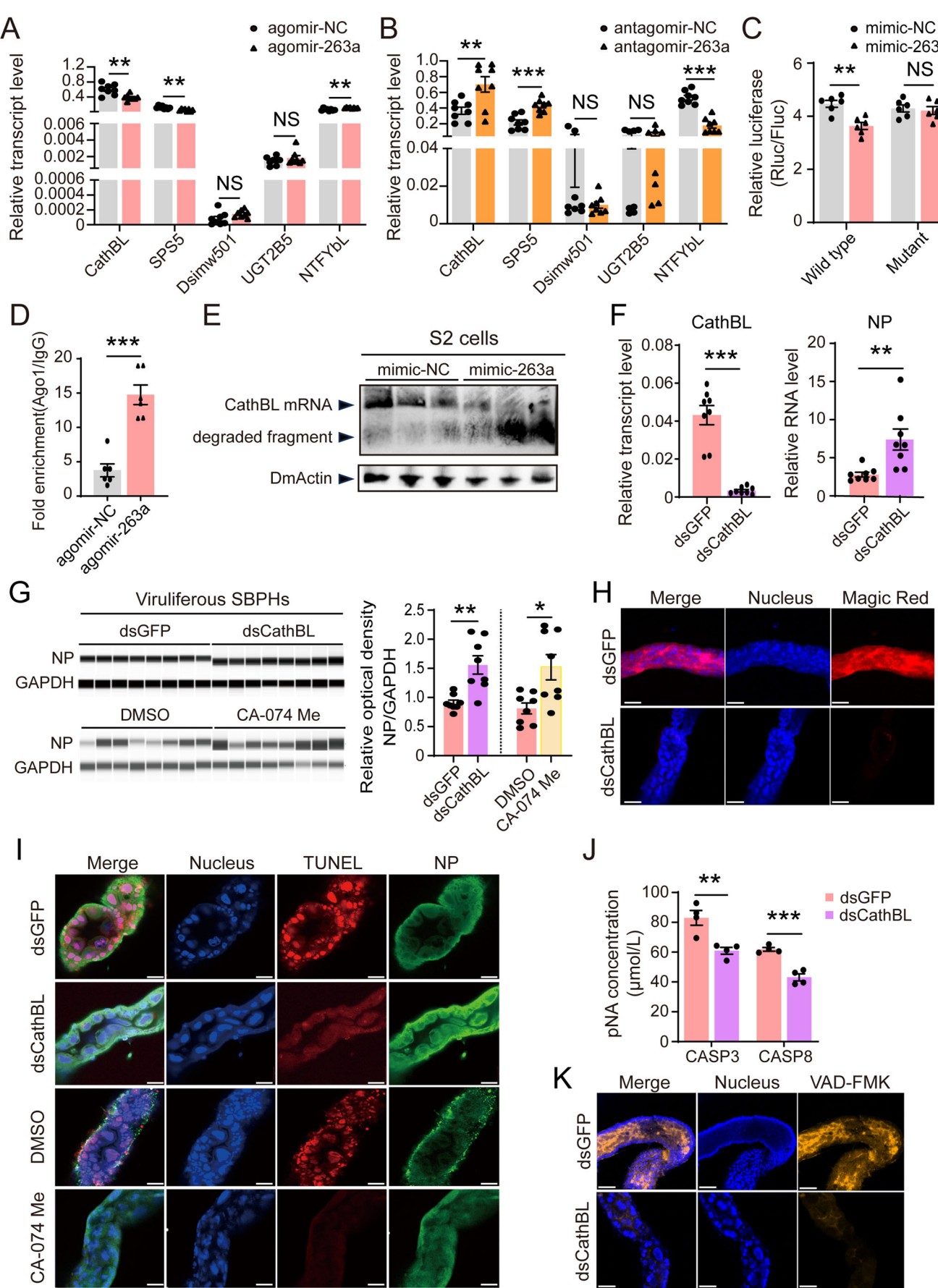

◄

**Figure 2. MiR-263a targets *Cathepsin B-like* (*CathBL*) to inhibit apoptosis for the favor of RSV accumulation in SBPH.**

(A, B) Relative transcript levels of the five candidate targets of miR-263a in nonviruliferous fourth-instar SBPH larvae after injected with agomir-263a/agomir-NC (A) or antagomir-263a/antagomir-NC (B) for 4 d. For (A), P values from left to right, $P = 0.0023$ (*CathBL*), $P = 0.0030$ (*SPS5*), $P = 0.0011$ (*NTFYbL*). For (B), P values from left to right, $P = 0.0080$ (*CathBL*), $P = 2.37E−04$ (*SPS5*), $P = 2.00E−06$ (*NTFYbL*). (C) Luciferase reporter assays in *Drosophila* S2 cells cotransfected with miR-263a/NC mimics and psiCHECK2 vectors containing wild type or mutant sequences of *CathBL* 5′UTR as the candidate target. $P = 0.0012$ (Wild type). (D) Relative enrichment of *CathBL* 5′ UTR in nonviruliferous SBPH measured by RIP combined with qPCR after injection with agomir-263a/agomir-NC for 4 d. The relative RNA level of *CathBL* 5′UTR, normalized to the IgG control, is presented as mean ± SE. $P = 7.30E−05$. (E) Northern blot analysis of *CathBL* mRNA levels in S2 cells following co-transfection with miR-263a/NC mimics and pAC5.1 vectors containing the 5′UTR and CDS of *CathBL*. Biotin-labeled probes specific for *CathBL* and the *Drosophila Actin* (*DmActin*) were utilized. (F) Relative transcript level of *CathBL* and the RNA level of viral *NP* in viruliferous fourth-instar larvae after injected with ds*CathBL*/ds*GFP*-RNA for 4 d. P values from left to right, $P = 9.05E−05$, $P = 0.0057$. (G) Accumulation of viral NP proteins in viruliferous SBPH after injection with ds*CathBL*/ds*GFP*-RNA, or Cathepsin B inhibitor CA-074 Me/DMSO for 4 d. NP and GAPDH as the reference protein were detected by JESS automated Western Blot system using a homemade monoclonal anti-NP antibody and a homemade polyclonal anti-GAPDH antibody, respectively. The relative optical densities of NP to that of GAPDH were calculated. P values from left to right, $P = 0.0041$, $P = 0.0139$. (H) Confocal images of midgut epithelium cells of SBPH after injection with ds*CathBL*/ds*GFP*-RNA for 4 d and treated with Magic Red. The blue signal is the nuclei stained with Hoechst. The red signal is from Magic Red label. Scale bars: 50 μm. (I) Immunohistochemistry for the midguts of viruliferous fourth-instar larvae at 4 d after injection of ds*CathBL*/ds*GFP*-RNA, or CA-074 Me/DMSO. The green signal is from an Alexa Fluor 488-labeled anti-NP monoclonal antibody. The red signal is from TUNEL label. The blue signal is the nuclei stained with Hoechst. Scale bars: 50 μm. (J) Caspase activities of viruliferous fourth-instar larvae after injected with dsCathBL/dsGFP-RNA for 4 d using kits for measuring human caspase 3 and 8 (CASP3 and CASP8). P values from left to right, $P = 0.0068$, $P = 4.00E−04$. (K) Immunohistochemistry for the midguts of viruliferous fourth-instar larvae at 4 d after injection of ds*CathBL*/ds*GFP*-RNA. The orange signal labeled by VAD-FMK showing caspase activity in vivo. The blue signal is the nuclei stained with Hoechst. Scale bars: 50 μm. For (A–G) and (J), three to eight replicates are shown for each group. Data are shown as mean ± SE. Student's t-tests were performed to evaluate the differences between two means at the same time point. NS, no significant difference. **$P < 0.01$. ***$P < 0.001$. Source data are available online for this figure.

PP716112). Mammalian Cathepsin B belongs to a class of lysosomal cysteine proteases of the papain family, which can induce apoptosis (Chwieralski et al, 2006; Sendler et al, 2016). We first tested the Cathepsin B proteinase activity of CathBL in midgut epithelial cells of SBPH using the magic red Cathepsin B substrate. After interfering the expression of *CathBL* with injection of ds*CathBL*-RNA, the total enzymatic activity reduced dramatically (Fig. 2H), showing that CathBL functioned as a typical Cathepsin B. Then we examined the potential role of CathBL in the apoptosis using a TUNEL assay. TUNEL-positive signals, indicative of apoptosis, significantly decreased in viruliferous SBPH midgut cells following dsCathB-RNA injection and more RSV accumulated in the cells compared to the injection of dsGFP-RNA (Fig. 2I). When CA-074 methyl ester, a specific inhibitor of Cathepsin B, was applied, the apoptotic reaction was also inhibited and the viral amounts in the gut and whole body enhanced accordingly (Fig. 2G,I). Furthermore, the impact of CathBL on caspase activities in SBPH was evaluated both in vitro and in vivo. We have successfully quantified SBPH caspase activities using kits typically applied for measuring human caspase 3 and 8 (Zhao et al, 2022b). Knockdown of *CathBL* reduced in vitro caspase activities of human caspase 3 and 8 (Fig. 2J), and SR-VAD-FMK-labeled caspase activities were notably reduced in midgut epithelial cells (Fig. 2K). These results demonstrated that CathBL functions as Cathepsin B proteinase to induce a caspase-dependent apoptosis, which is unfavorable for RSV accumulation in SBPH.

## Mature miR-263a is secreted in rice by SBPH

Given prior evidence of miR-263a in the saliva of mosquitoes and ticks (Arca et al, 2019), we tested the possibility of mature miR-263a secreted from SBPH to rice. Continuous feeding by nonviruliferous SBPH for a week led to the detection of mature miR-263a in rice leaves, confirmed by northern blot, qPCR, and Sanger sequencing (Fig. 3A,B). When nonviruliferous SBPHs were allowed to feed on rice leaves 2 d, the amount of miR-263a in rice kept constant within 6 d while nearly undetectable at 12 d without SBPH infestation (Fig. 3C). Although the expression of miR-263a

was downregulated in the salivary glands of SBPH upon RSV infection (Fig. 1B), the amounts of miR-263a secreted in rice were comparable between nonviruliferous and viruliferous SBPHs with continual infestation for 2, 4, or 6 d (Fig. 3D).

The 3′-end of plant miRNAs has a 2′-O-methylation, which makes miRNAs resistant to RNase or oxidation (Yu et al, 2005). In contrast, mammal and insect miRNAs are typically not methylated (Grimson et al, 2008; Hartig and Förstemann, 2011). To examine miR-263a methylation in rice, we performed oxidation and β-elimination treatment on RNAs extracted from the rice leaves fed on by SBPH. No mobility shift of miR-263a was observed in rice after the treatment whereas miR-263a from insects or the in vitro synthesized miR-263a with 3′-OH (miR-263a-OH) showed a noticeable mobility shift of approximately 1.5 nucleotides (Fig. 3E). This indicated that SBPH miR-263a is methylated in rice. Based on the sequence characteristics of miR-263a, we supposed that miR-263a could be incorporated in rice Argonaute (Ago) 4, which predominantly recruits 24-nt miRNAs with a 5′-terminal adenine and functions mainly in the nucleus (Campo et al, 2021; Ye et al, 2012). A RIP-qPCR assay was carried out on the cytoplasmic and nuclear fractions of rice fed on by SBPHs, using commercial polyclonal antibodies against rice Ago4 and Ago1. It was found that miR-263a was significantly enriched in the Ago4-immunoprecipitation from the nuclear fraction but not cytoplasm compared to the IgG immunoprecipitation and miR-263a was not obviously associated with Ago1, which prefers to recruit 21-nt miRNAs (Fig. 3F). These results demonstrated that miR-263a is imported in rice nucleus and bound to Ago4 in a methylated status.

## MiR-263a inhibits RSV accumulation in rice

To clarify the role of miR-263a in RSV infection in rice, we initiated our study by injecting miR-263a agomir in viruliferous SBPHs to facilitate increased miR-263a secretion into in rice leaves (Fig. 4A). Consequently, a significant decrease in the RNA and protein level of RSV NP in rice was observed compared to the injection of control agomir (Fig. 4B). Subsequently, we engineered a transient over-expression strategy for miR-263a in rice protoplasts by using the

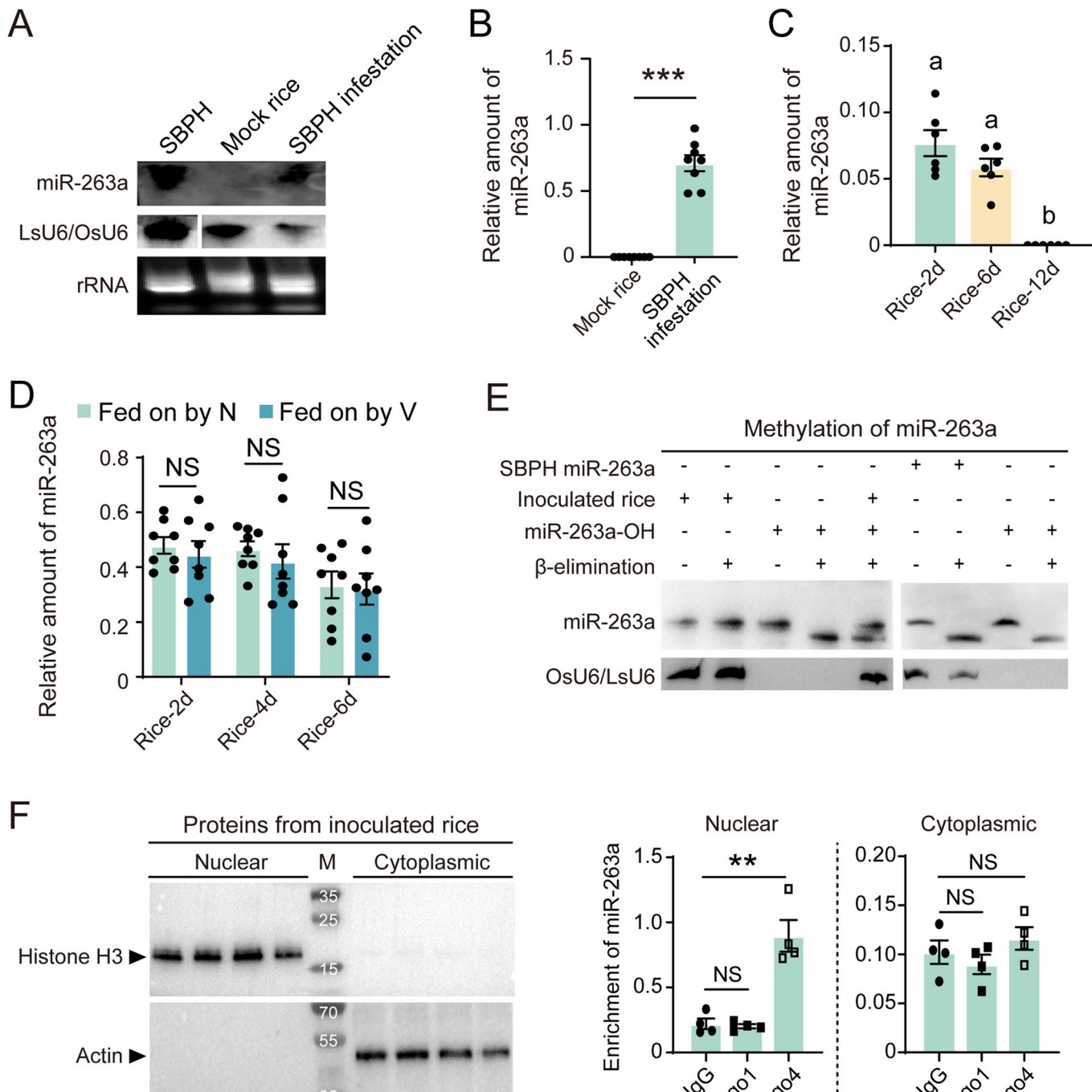

precursor stem-loop structure of rice miR-1876, which generates a 24-nt mature miRNA in rice (Wu et al, 2010). An overexpression plasmid (OE263a) was designed for boosting miR-263a levels, achieved by replacing the miRNA and miRNA* sequences of rice miR-1876 with those of miR-263a and its complementary strand, miR-263a* (Fig. 4C). After co-transfection of RSV crude extracts from infected rice and the OE263a plasmids into protoplasts for 48 h, miR-263a was expressed and the RNA level of RSV *NP* dramatically decreased in contrast to the transfection of empty plasmid (Fig. 4D). Stable transgenic lines harboring OE263a were further generated, in which both precursor

and mature miR-263a were successfully detected (Fig. 4E,F). Compared to the wild-type (WT) rice, the transgenic lines displayed lower plant height, smaller 1000-grain weight and grain width (Fig. EV3A,C,E). Following a 7-d feeding period by viruliferous SBPHs, OE263a lines showed lower viral loads as compared to WT rice (Fig. 4G), and the disease incidence was also lower under greenhouse conditions, with only 52% in the OE263a line versus 75% in the WT within 30 d post inoculation (Figs. 4H and EV3F). These results demonstrated that miR-263a exerts an antiviral effect against RSV in rice.

**Figure 3.  Mature miR-263a is secreted in rice by SBPH.**

(A) Northern blot analysis for determining the existence of mature miR-263a in nonviruliferous SBPH, healthy rice leaves, and rice constantly fed on by nonviruliferous fourth-instar SBPH larvae for 7 d. The signal was detected using a biotin-labeled LNA oligonucleotide probe for miR-263a. LsU6 snRNA and OsU6 snRNA serve as internal references for SBPH and rice, respectively, and rRNA demonstrates comparable initial loading amounts across different samples. (B) qPCR analysis for the amount of miR-263a relative to that of OsU6 snRNA in healthy rice leaves and rice leaves continuously fed on by nonviruliferous fourth-instar SBPH larvae for 7 d. $P = 7.34\text{E}{-}06$. (C) qPCR analysis for the amount of miR-263a relative to that of OsU6 snRNA in rice leaves following a 2-d feeding period by nonviruliferous fourth-instar SBPHs, or with additional 4 and 10 d post-removal of insects. Data are shown as mean ± SE. Comparisons among multiple groups were conducted using one-way analysis of variance (ANOVA) followed by Tukey's test. Different letters indicate significant differences. (D) qPCR analysis for the amount of miR-263a relative to that of OsU6 snRNA in rice leaves fed on by nonviruliferous and viruliferous fourth-instar larvae with continual infestation for 2, 4, or 6 d. For (B), (C), and (D), each rice leaf was inoculated with 10 insects in a microcage, and six to eight replicates with two leaves per replicate were collected. (E) β-elimination analysis of miR-263a in rice following feeding by nonviruliferous fourth-instar SBPH larvae, artificially synthesized miR-263a harboring free hydroxyl groups at the 3′ terminal ribose, and miR-263a from nonviruliferous SBPH larvae. RNA samples were treated (+) or untreated (−) with the chemicals for β-elimination. OsU6 snRNA and LsU6 snRNA serve as internal references for rice and SBPH, respectively. In Lane 5, RNAs extracted from rice fed on by SBPH and synthesized miR-263a-OH standards were mixed at the ratio of 1:1. (F) Enrichment of miR-263a detected by RIP-qPCR analysis using the anti-Ago1 and anti-Ago4 antibodies in the nuclear and cytoplasmic fractions of rice leaves following feeding by nonviruliferous fourth-instar SBPH larvae. qPCR was performed to quantify the enrichment of miR-263a in the IP fraction compared to the Input. Four biological replicates were prepared. $P = 0.0019$. For (B), (D), and (F), values are shown as mean ± SE and were compared using Student's t test. NS, no significant difference. **$P < 0.01$. ***$P < 0.001$. Source data are available online for this figure.

## MiR-263a upregulates *GATA transcription factor 19* to resist RSV in rice

Candidate target genes of miR-263a in rice were predicted using the psRNATarget website with rice genomic data. Seventeen targets were screened, including 7 untranslated regions, 6 exons, 4 introns, and 2 regions situated in intergenic areas (Table EV2). qPCR analysis of the genes hosting these 17 target sites revealed that the transcript level of *GATA transcription factor 19* (*GATA19*, Os03g52450.1) was upregulated and *retrotransposon* (Os07g11600.1) was downregulated in OE263a lines compared to WT, while the rest 15 genes remained unchanged (Figs. 5A and EV4). Predicted miR-263a target sites were mapped to the first intron of *GATA19* and an exon for *retrotransposon* (Fig. 5B and Table EV2). To validate the precise cleavage sites, 5′ RNA ligase-mediated rapid amplification of cDNA ends (5′RLM-RACE) was applied on OE263a lines. Sequencing confirmed in vivo cleavage between the 11th and 12th base pair of the miR-263a target site in *GATA19* (Fig. 5B), while no cleavage was detected in the predicted target site of *retrotransposon*. Measuring *GATA19*'s primary transcript by using a primer at the first intron, lower amounts were detected in OE263a lines than WT, implicating *GATA19* as a prime miR-263a target (Fig. 5A). We further verified the direct interaction between miR-263a and the intron target sequence of *GATA19* which was inserted behind the stop codon of *EGFP* (*EGFP*-target), using EGFP reporter expression system in rice protoplasts. The green fluorescence of EGFP was weaker when miR-263a and *EGFP*-target were overexpressed together than the *EGFP*-target over-expressed singly (Fig. 5C), suggesting that miR-263a is capable of cleaving the intron target sequence of *GATA19*, leading to a subsequent reduction in EGFP protein levels. Overexpression of miR-263a with a plasmid harboring *GATA19*'s first two exons and first intron (*Ex-In-Ex*) increased the exon transcripts in rice protoplasts (Fig. 5D). Mutations at the 17-nt binding site within the intron region (*Ex-InMT-Ex*) abolished the activation effect of miR-263a on *GATA19* exons (Fig. 5D). To explore the effect of GATA19 on RSV infection, *GATA19* knockout mutant rice was generated by CRISPR/Cas9 technology. Two sgRNAs were designed within the first and second exon. Two independent mutant lines were obtained, harboring a "T" insertion in sgRNA1 and a "AC" deletion in sgRNA2 target site for mutant line 1, and a "GG"

deletion in sgRNA1 and a "C" insertion in sgRNA2 target site for mutant line 2 (Fig. 5B). Western blot confirmed the absence of GATA19 in mutant lines using the homemade anti-GATA19 polyclonal antibody (Fig. 5E). Interestingly, the molecular weight of GATA19 detected in WT rice was larger than 29 kDa, the putative size estimated from 816 bp of open reading frame, probably due to post-translational modifications (Fig. 5E). *GATA19* loss-of-function had no impact on plant height, 1000-grain weight or grain width, but enhanced grain length under greenhouse conditions (Fig. EV3B,D,E). After fed on by viruliferous SBPHs for 7 d, mutant lines accumulated significantly more RSV (Fig. 5F) and showed slightly higher disease incidences than WT within 30 d post inoculation (Figs. 5G and EV3F), highlighting GATA19's antiviral function and miR-263a's role in enhancing *GATA19* expression to confer RSV resistance in rice.

## GATA19 positively regulates jasmonate signaling pathway

As a transcription factor, understanding the antiviral mechanism of GATA19 deserved further exploration. Given its membership in the TIFY family, which includes pivotal JAZ proteins known to interact with transcription factor MYC2 and inhibit the activation of the Jasmonate (JA) signaling pathway (Chini et al, 2007), the potential role of GATA19 in modulating this interaction was of interest, particularly since the JA pathway mediates the major resistance to RSV infection in rice (Wu et al, 2024).

To verify this hypothesis, we measured the contents of JA and its derivative jasmonoyl-L-isoleucine (JA-Ile) in WT, OE263a, and *GATA19*KO lines. Elevated *GATA19* in OE263a lines corresponded with increased JA and JA-Ile levels, whereas they were reduced in *GATA19*KO mutants compared to WT (Fig. 6A). For the 12 genes involved in JA synthesis and JA signal transduction, the transcript levels of *LOX1*, *LOX2*, *AOS2*, *JAZ8*, *JAZ12*, *MYC2*, and *JMT1* were upregulated in OE263a lines and *LOX1*, *LOX2*, *AOS2*, *JAZ8*, and *JAMYB* were downregulated in *GATA19*KO lines (Fig. EV5). To assess whether the amount of miR-263a injected by insects is sufficient to alter the *GATA19* transcript levels and influence downstream jasmonic acid (JA) signaling, we measured *GATA19* transcript levels, JA and JA-Ile contents in rice plants fed upon by nonviruliferous SBPHs with injection of miR-263a antagomir or control antagomir. The treatment

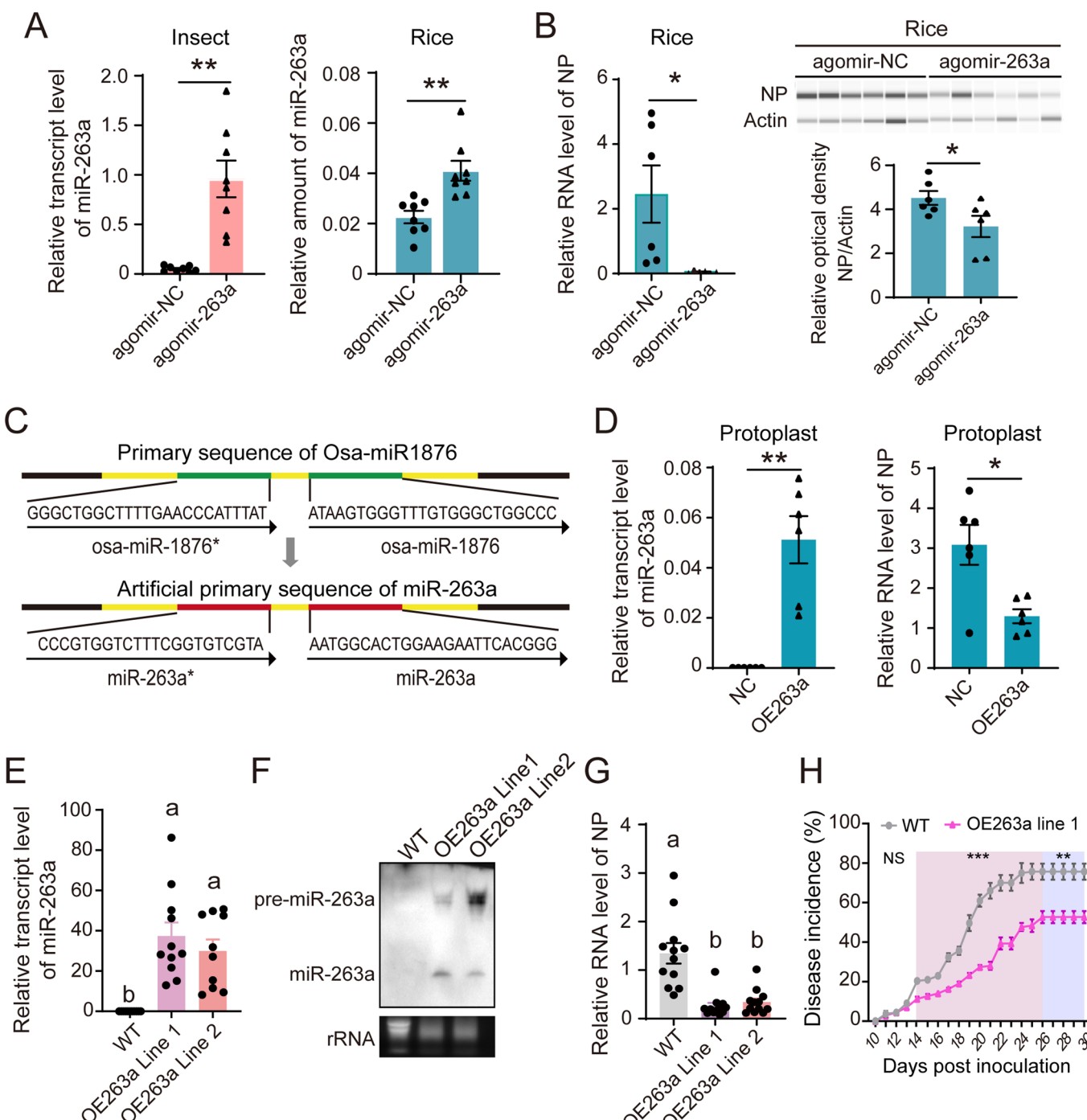

with the miR-263a antagomir effectively reduced the amount of miR-263a secreted into the rice plants (Fig. EV6). Correspondingly, *GATA19* transcript levels and JA content decreased significantly whereas JA-Ile content remained unchanged (Figs. 6B and EV6). These findings indicate that miR-263a induces *GATA19* expression, which subsequently activates the JA signaling pathway in rice.

Previous studies have demonstrated direct interactions between rice JAZ1, JAZ8, and JAZ10 with MYC2 (Uji et al, 2016). Our Co-IP assays showed that the recombinantly expressed JAZ1-His could pull down GATA19-V5 while JAZ8-His or JAZ10-His did not

(Fig. 6C). Furthermore, JAZ1-His successfully captured endogenous GATA19 from rice nuclear extracts (Fig. 6D), and a BIFC assay in *Nicotiana benthamiana* epidermal cells confirmed their nuclear interaction (Fig. 6E). As supposed, GATA19-V5 did not pull down MYC2-Flag in vitro due to the lack of a Jas domain (Fig. 6F). Interestingly, in a competitive binding setup, while JAZ1-His could bind MYC2-Flag, an increase in GATA19-V5 reduced MYC2-Flag pulled down by JAZ1-His (Fig. 6G), suggesting that the upregulated expression of *GATA19* could weaken the binding of JAZ1 with MYC2, thus activating the JA pathway.

◄

**Figure 4.  MiR-263a inhibits RSV accumulation in rice.**

(A) Relative transcript level of miR-263a in viruliferous fourth-instar SBPH larvae after injection with agomir-263a/agomir-NC for 4 d and in rice fed on by these treated insects. LsU6 snRNA served as an internal control for SBPH, while OsU6 snRNA was utilized for rice. P values from left to right, $P = 0.0017$, $P = 0.0015$. (B) Relative RNA and protein level of viral NP in rice after 7 d of feeding by insects treated as described in (A). Each leaf was inoculated with 10 insects trapped in a microcage, and two leaves were collected as one replicate. For qPCR, OsU6 snRNA served as the internal control for miR-263a normalization, and *UBQ10* was used for viral *NP* quantification. Western blot analysis, automated by the JESS system, employed a homemade monoclonal anti-NP antibody to detect NP and a monoclonal anti-Actin antibody for plant Actin, which acted as the loading control. The ratio of NP to Actin signal intensities was calculated from six independent replicates. P values from left to right, $P = 0.0449$, $P = 0.049$. (C) Diagram illustrating the design of an artificial primary of miR-263a for generating overexpression vectors. Here, the native miR-1876ro and miR-1876* sequences (highlighted in green) are substituted with miR-263a and miR-263a* sequences (indicated in red). The black arrow denotes the nucleotide orientation. (D) Relative transcript level of miR-263a and the RNA level of viral *NP* in rice protoplasts cotransfected with plant-derived RSV crude extractions and OE263a plasmids for 48 h. P values from left to right, $P = 0.0029$, $P = 0.0142$. (E) Relative transcript level of miR-263a in wild-type (WT) and OE263a transgenic lines. (F) Identification of mature and processor of miR-263a in WT and OE263a rice lines by Northern blot. Biotin-labeled LNA oligonucleotide probes against miR-263a were used to detect miR-263a. rRNA was used to represent the quantity of total RNAs utilized. (G) Relative RNA level of viral *NP* in WT and OE263a rice lines after feeding by viruliferous SBPH for 7 d. (H) The disease incidence of WT and OE263a rice fed on by viruliferous SBPH for 7 d. Five rice seedlings per replicate and six replicates were applied. From 14 to 30 dpi, $P = 2.70E-04$ (14 dpi), $P = 7.00E-04$ (15 dpi), $P = 9.90E-05$ (16 dpi), $P = 2.13E-04$ (17 dpi), $P = 2.36E-04$ (18 dpi), $P = 4.09E-04$ (19 dpi), $P = 5.40E-05$ (20 dpi), $P = 1.24E-04$ (21 dpi), $P = 4.56E-04$ (22–23 dpi), $P = 9.44E-04$ (24 dpi), $P = 7.86E-04$ (25 dpi), $P = 0.0014$ (25–30 dpi), from left to right. For (E) and (G), data are shown as mean ± SE. Comparisons among multiple groups were conducted using one-way analysis of variance (ANOVA) followed by Tukey's test. Different letters indicate significant differences. For (A), (B), (D), and (H), values are shown as mean ± SE and were compared by Student's t test. *$P < 0.05$. **$P < 0.01$. ***$P < 0.001$. Source data are available online for this figure.

## Discussion

In this study, we revealed how insect microRNAs regulate arbovirus infection in insect vector and in plant host as cross-kingdom effectors. The insect conserved miR-263a inhibits Cathepsin B-mediated apoptosis to facilitate viral proliferation in midgut epithelial cells. However, the expression of miR-263a is in turn suppressed by RSV to control viral titer in insect vector. At the same time, miR-263a is delivered in rice with insect saliva secretion. As an exogenous microRNA, miR-263a upregulates the expression of transcription factor *GATA19* to hinder JAZ1 binding with MYC2, thus activating jasmonate signaling pathway to afford rice resistance to RSV (Fig. 7). Therefore, miR-263a plays an opposite role in viral infection between insect vector and plant host.

It is unusual to find that miR-263a cleaves the first intron of *GATA19* to promote the mature of *GATA19* transcript. RNA splicing is an essential regulatory step of generating mature mRNAs, in which introns are removed and exons are ligated (Will and Lührmann, 2011). sRNAs have been reported to target introns for facilitating RNA splicing in insect or human cells. For example, piRNAs target intronic branch point sequence of *NPF1* primary mRNA, preventing hairpin structure formation to enhance splicing efficiency in locust nuclei (Wang et al, 2022). Two artificially synthesized siRNAs targeting the intron 33 of human *fibronectin* gene affect alternative splicing by promoting a substantially greater inclusion-to-exclusion ratio of the exon 33 (Allo et al, 2009). Although numerous miRNAs are originated from introns (Kim and Kim, 2007; Melamed et al, 2013), there has been no direct evidence to demonstrate the effect of these miRNAs on RNA splicing through targeting introns. Based on degradome sequencing data, rice has 1912 miRNA-intron pairs, indicating the possibility of miRNA cleavage sites within the intron regions, and the miRNAs involved in intron-interaction are predominantly 22 and 24 nt (Meng et al, 2013). We proved that the 24 nt of miR-263a cleaves the first intron of *GATA19*, supporting the findings from rice degradome. Generally, the first intron is the longest and most conserved, and plays a particularly important regulatory role in transcription control (Majewski and Ott, 2002). Removal of the first intron of *DYN2* gene inhibits gene transcription (Furger et al,

2002). The order of intron excision frequently deviates from their sequential placement in genes or transcripts. In human cells, introns that are removed later tend to be slightly longer and have weaker splice sites (Choquet et al, 2023). We speculate that miR-263a could promote the splicing efficiency of the first intron to accelerate the maturation of *GATA19* transcript.

MiR-263a undergoes methylation to enhance the stability in rice. We found that miR-263a persisted in rice at least 6 days. The 2′-O-methylation of the 3′ terminal ribose has been observed in miRNAs and siRNAs in plants, and serves as a protective mechanism against 3′-5′ degradation and periodate of small RNAs (Horwich et al, 2007; Yu et al, 2005). HEN1 mediates 2′-O-methylation at the 3′ end of each strand of the miRNA/miRNA* duplex and siRNA duplexes in the nucleus while single-stranded small RNAs and pre-miRNAs fail to be methylated by HEN1 (Yang et al, 2006). In our study, we observed miR-263a loading onto Ago4 within rice nuclei. This suggests that miR-263a might be methylated by HEN1 in the form of miRNA/miRNA* duplex before binding to intronic target sites.

Our work unveiled a new mechanism for GATA transcription factors to regulate the JA pathway. Both GATA19 and JAZ proteins belong to TIFY family but in different subfamilies (Bai et al, 2011). The largest difference between GATA19 and JAZ is that GATA19 lacks of the Jas domain, a critical component for the interaction between JAZ and MYC2 (Chini et al, 2007). This explains why GATA19 does not directly bind MYC2. JAZ functions as negative regulator of JA pathway by binding to MYC2. The homodimers or heterodimers of JAZ significantly enhance the repressive activity in *Arabidopsis* (Yamada et al, 2012). Interestingly, when lacking of Jas domain, JAZ proteins are prone to form heterodimers (Chini et al, 2009). Therefore, GATA19 is well-suited for forming heterodimers with JAZ1. However, the binding of GATA19 with JAZ1 does not produce the upgraded repressive effect on the JA pathway as JAZ homodimers do. Instead, GATA19 retards the interaction between JAZ1 and MYC2, ultimately activating the JA pathway. Therefore, GATA19 and JAZ play opposite roles in regulating the JA pathway.

As a double agent, miR-263a produces profound influences on the virus-vector-host tripartite relation. In insect vector, miR-263a facilitates RSV accumulation through suppressing apoptotic

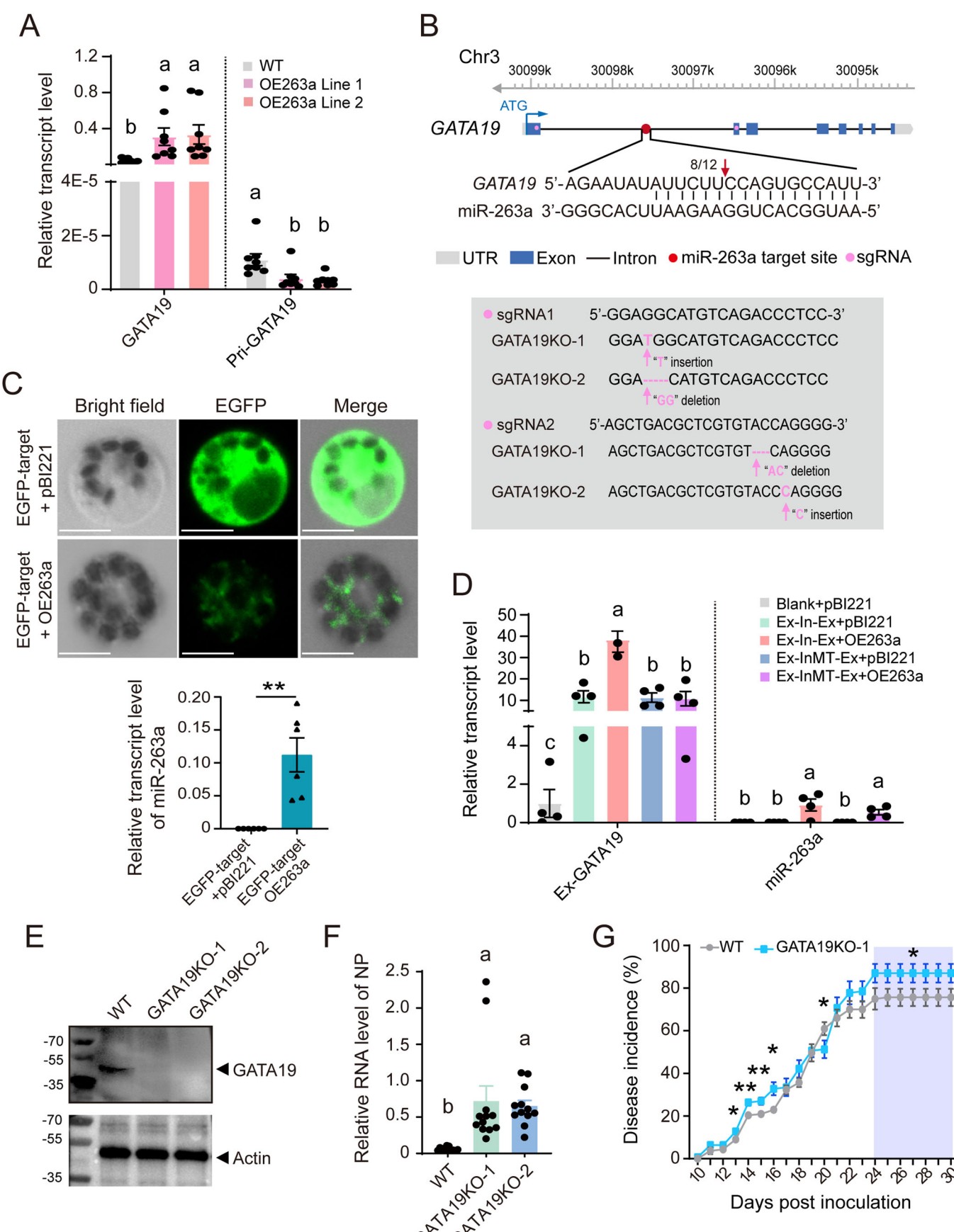

**Figure 5. MiR-263a upregulates *GATA transcription factor 19* to resist RSV in rice.**

(A) Relative transcript level of *GATA19* and its primary transcript (Pri-GATA19) in WT and OE263a rice lines. (B) Generation of *GATA19* knockout (*GATA19*KO) mutant rice lines via CRISPR/Cas9: On the top, the placement of two sgRNAs (denoted by pink circles) is illustrated within *GATA19*'s open reading frame. The red arrow signifies the experimentally validated splicing site, confirmed through 5'RLM-RACE and sequencing, with the adjacent number reflecting the frequency of 5'RLM-RACE products cleaved at that specific site. Turning to the bottom, the diagram showcases the two sgRNA sequences that specifically *target GATA19*, with a pink arrow pinpointing the site of induced mutation. (C) miRNA target-GFP reporter expression assay for the interaction of miR-263a and the target site within the first intron of *GATA19*. Rice protoplasts were cotransfected with pBI-EGFP-target and pBI-OE263a, or cotransfected with pBI-EGFP-target and pBI221 empty vectors. Images were captured at 36 h post-transfection. Scale bars: 5 μm. Relative transcript level of miR-263a in rice protoplasts was detected by qPCR. $P = 0.0065$. (D) Relative transcript levels of mRNA from an artificial primary transcript of *GATA19* and miR-263a measured in rice protoplasts. Protoplasts were transfected with a recombinant pBI221 plasmid containing the first two exons and the intervening intron of *GATA19* (*Ex-In-Ex*) or a mutant plasmid with a 17-nucleotide mutated target site within the intron (*Ex-InMT-Ex*). Transfections were conducted either in combination with the miR-263a overexpression vector (OE263a) or with the empty pBI221 vector as a control. (E) Protein analysis of GATA19 in WT and *GATA19*KO mutant lines using Western blot assays. NP was detected using a homemade monoclonal anti-NP antibody. Plant Actin that was detected using a monoclonal anti-Actin antibody was used as the reference protein. The relative optical densities of NP to that of Actin were calculated. (F) Relative RNA level of viral *NP* in *GATA19*KO mutant lines compared to WT post inoculation with viruliferous fourth-instar SBPH larvae. (G) The disease incidence of WT and *GATA19*KO rice fed viruliferous fourth-instar SBPH larvae for 7 d. Five rice seedlings per replicate and six replicates were applied. $P = 0.0100$ (13 dpi), $P = 0.0031$ (14 dpi), $P = 0.0065$ (15 dpi), $P = 0.0058$ (16 dpi), $P = 0.0310$ (20 dpi), $P = 0.0351$ (24 dpi), $P = 0.0299$ (25–30 dpi), from left to right. For (A), (D), and (F), data are shown as mean ± SE. Comparisons among multiple groups were conducted using one-way analysis of variance (ANOVA) followed by Tukey's test. Different letters indicate significant differences. For (C) and (G), values are shown as mean ± SE and were compared by Student's t test. *$P < 0.05$. **$P < 0.01$. Source data are available online for this figure.

reaction as well as binding to the extended terminal of viral genome (Zhao et al, 2021). However, the expression of miR-263a is downregulated by viral NP and a virus-derived small RNA of RSV (Zhao et al, 2021). This interaction between miR-263a and RSV in insect vector secures a limited viral load, which is critical for keeping insect vector healthy and competent. When miR-263a is secreted in rice, it increases JA content. It has been reported that the RSV NP triggers JA-AGO18-mediated immune defense in rice (Yang et al, 2020). Knockout of *MYC2* renders rice more susceptible to RSV (Hu et al, 2020). RSV NP also activates plant defenses against the virus by inducing JA accumulation but at the same it attracts SBPH for feeding, thereby facilitating viral transmission (Han et al, 2020). Overall, as insect effector, miR-263a induces an antiviral reaction in rice cells but may simultaneously benefit for viral transmission.

# Methods

### Reagents and tools table

| Reagent/Resource | Reference or Source | Identifier or Catalog Number |
|---|---|---|
| **Experimental models** | | |
| Rice stripe virus Jiangsu isolate (JSHA) | Zhao et al, 2018 | N/A |
| SBPH populations | This paper | N/A |
| *O. sativa* subsp. *japonica* wuyujing3 | This paper | N/A |
| *O. sativa* subsp. *japonica* cv. Nipponbare Wild-type | This paper | N/A |
| *O. sativa* subsp. *japonica* cv. Nipponbare OE263a | This paper | N/A |
| *O. sativa* subsp. *japonica* cv. Nipponbare GATA19KO | This paper | N/A |
| *Rice protoplasts* | This paper | N/A |

| Reagent/Resource | Reference or Source | Identifier or Catalog Number |
|---|---|---|
| *Nicotiana benthamiana* | This paper | N/A |
| GV3101 Agrobacterium Competent Cells | Zoman | Cat# ZK295 |
| *Drosophila* S2 cells | Thermo Scientific | Cat# R69007 |
| **Recombinant DNA** | | |
| psiCHECK2 | Promega | Cat# C8021 |
| pciCHECK2-CathBL-5'UTR | This paper | N/A |
| pciCHECK2-SPS5-5'UTR | This paper | N/A |
| pciCHECK2-NTFYbL-3'UTR | This paper | N/A |
| pAC5.1B | Invitrogen | Cat# V411020 |
| pAC5.1B-CathBL | This paper | N/A |
| pBI221 | MiaoLing | Cat# P6700 |
| pBI221-EGFP | HonorGene | Cat# VZC0336 |
| pBI221-EGFP-targe | This paper | N/A |
| pBI-OE263a | This paper | N/A |
| pBI-Ex-In-Ex | This paper | N/A |
| pBI-Ex-InMT-Ex | This paper | N/A |
| **Antibodies** | | |
| Anti-GATA19 polyclonal antibody | This paper | N/A |
| Anti-RSV NP monoclonal antibody | Zhao et al, 2016b | N/A |
| Anti-Ago1 monoclonal antibody | Zhao et al, 2019a | N/A |
| Anti-GAPDH polyclonal antibody | Wang et al, 2013 | N/A |
| Anti-Histone H3 Mouse Monoclonal Antibody | EASYBIO | Cat# BE3021 |

| Reagent/Resource | Reference or Source | Identifier or Catalog Number |
|---|---|---|
| Anti-plant-Actin Mouse Monoclonal Antibody | EASYBIO | Cat# BE0028 |
| Anti-V5-Tag Mouse Monoclonal Antibody | EASYBIO | Cat# BE2032 |
| Anti-His-Tag Mouse Monoclonal Antibody | EASYBIO | Cat# BE2019 |
| Goat Anti-Mouse IgG (H&L)-HRP | EASYBIO | Cat# BE0102 |
| Goat Anti-Rabbit IgG (H&L)-HRP | EASYBIO | Cat# BE0101 |
| Goat anti-Mouse IgG (Fc specific, Heavy Chain), HRP | EASYBIO | Cat# BE0104 |
| Goat anti-Rabbit IgG (Fc specific, Heavy Chain), HRP | EASYBIO | Cat# BE0106 |
| Ago1/Anti-Argonaute 1 Antibody | PhytoAB | Cat# PHY1381S |
| Ago4/ Anti-Argonaute 4 Antibody | PhytoAB | Cat# PHY0202S |
| Recombinant Anti-DDDDK tag (Binds to FLAG® tag sequence) antibody | Abcam | Cat# ab205606 |
| Goat Anti-Mouse IgG H&L | Abcam | Cat# ab6708 |
| Streptavidin, HRP labeled | Biodragon | Cat# BF06185X |
| Alexa Fluor 488 conjugated secondary antibody | Invitrogen | Cat# A-11008 |
| **Oligonucleotides and other sequence-based reagents** | | |
| Primers and probes | This paper | Appendix Table S1 |
| **Chemicals, Enzymes and other reagents** | | |
| Caspase inhibitor Z-VAD-FMK | Beyotime | Cat# C1202-0.02ml |
| CA-074 methyl ester | ChemeGen | Cat# 147859-80-1 |
| Chloroform | Sigma-Aldrich | Cat# C2432-500M |
| Dimethyl sulfoxide (DMSO) | Sigma-Aldrich | Cat# D2650 |
| Protease Inhibitor Cocktail | MilliporeSigma | Cat# P8849 |
| M-MLV Reverse Transcriptase | Promega | Cat# PAM1701 |
| Oligo(dT)15 Primer | Promega | Cat# PAC1101 |
| Random Pimers | Promega | Cat# PAC1181 |
| Ampicillin | Sangon | Cat# B541011 |
| BrightStar™-Plus Positively Charged Nylon Membrane | Invitrogen | Cat# AM10102 |
| Glycogen (5 mg mL$^{-1}$) | Invitrogen | Cat# AM9510 |
| Lipofectamine™ 3000 Transfection Reagent | Invitrogen | Cat# 11668019 |

| Reagent/Resource | Reference or Source | Identifier or Catalog Number |
|---|---|---|
| North2South™ Hybridization Stringency Wash Buffer (2×) | Thermo Scientific | Cat# 37555 |
| Nucleic Acid Detection Blocking Buffer | Thermo Scientific | Cat# 89880A |
| Substrate Equilibration Buffer | Thermo Scientific | Cat# 89880C |
| TRIzol™ Reagent | Invitrogen | Cat# 15596026 |
| UltraPure™ DNase/RNase-Free Distilled Water | Invitrogen | Cat# 10977015 |
| UltraPure™ SSC, 20X | Invitrogen | Cat# 15557036 |
| 2×RNA Loading Dye | NEB | Cat# B0363S |
| T4 DNA Ligase | NEB | Cat# M0202S |
| T4 ligase reaction buffer | NEB | Cat# B0216S |
| dNTPs (2.5 mM each) | Tiangen | Cat# CD117-02 |
| 10×TBE buffer | Solarbio | Cat# T1051 |
| Acid phenol chloroform | Merck | Cat# 77619 |
| Sodium periodate, AR, 99.5% | Macklin | Cat# S817518 |
| Borate Buffer | Macklin | Cat# B885515 |
| Super Hybridization Solution | Zeyebio | Cat# ZY7352PD |
| Nucleic Acid Pre-Hybridization Solution | Zeyebio | Cat# ZY601922B |
| FDA (10 mg ml$^{-1}$) | Coolaber | Cat# PPT5351 |
| 4% Paraformaldehyde Fix Solution (PFA) | BBI Life Science | Cat# E672002-0100 |
| VAD-FMK | Abcam | Cat# ab270770 |
| Caspase 3 Activity Assay Kit | Beyotime | Cat# C1116 |
| Caspase 8 Activity Assay Kit | Beyotime | Cat# C1152 |
| One-Step TUNEL Apoptosis Assay Kit (Red Fluorescence) | Beyotime | Cat# C1089 |
| Cathepsin B Assay Kit (Magic Red) | Abcam | Cat# Ab270772 |
| Dual-Glo® Luciferase Assay System | Promega | Cat# E2920 |
| Dynabeads Protein G | Invitrogen | Cat# 10003D |
| T7 RiboMAX™ Express RNAi System | Promega | Cat# P1700 |
| KOD-Plus-Mutagenesis Kit | Toyobo | Cat# SMK-101 |
| miRcute Plus miRNA First-Strand cDNA Kit | Tiangen | Cat# 4992909 |
| miRcute Plus miRNA qPCR Kit (SYBR Green) | Tiangen | Cat# 4992779 |

| Reagent/Resource | Reference or Source | Identifier or Catalog Number |
|---|---|---|
| Premium One Cell-Free Protein Expression Kit | Tiangen | Cat# CFS-EDX-ONE |
| In-Fusion HD Cloning kit | Takara | Cat# 638943 |
| FirstChoice™ RLM-RACE | Invitrogen | Cat# AM1700M |
| pEASY®-Blunt Zero Cloning Kit | TRANS | Cat# CB501-01 |
| Plant nucleus and cytoplasm extraction kit | Bestbio | Cat# BB-31124 |
| Rice Protoplast Preparation and Transformation Kit | Coolaber | Cat# PPT111-5T |
| SuperSignal West Femto | Thermo Scientific | Cat# 34094 |
| Superscript III First-Strand Synthesis System | Thermo Scientific | Cat# 18080051 |
| Magna RIP RNA-binding protein immunoprecipitation kit | Merck | Cat# 17-700 |
| **Software** | | |
| GraphPad Prism 9.0 | https://www.graphpad.com/ | N/A |
| SPSS statistics 19.0 | https://spss.software.informer.com/download/ | N/A |
| ImageJ | https://imagej.net/ij/ | N/A |
| LightCycler480 | https://lightcycler-961.software.informer.com/download/ | N/A |
| miRanda | http://www.microrna.org/microrna | N/A |
| RNAhybrid | https://bibiserv.cebitec.uni-ibielefeld.de/rnahybrid | N/A |
| psRNATarget | http://plantgrn.noble.org/psRNATarget/ | N/A |
| **Other** | | |
| RNA-seq reads (Full-length transcriptome of SBPH) | Tong et al, 2022 | PRJCA005421 |
| RNA-seq reads (Gut transcriptom of SBPH) | Zhao et al, 2016a | SRP048969 |
| Oryza sativa (rice) genomic DNA data | http://rice.uga.edu/analyses_search_blast.shtml | N/A |

## Virus, insect, and plant preparation

Viruliferous and nonviruliferous SBPH populations were incubated separately on 2–3 cm seedlings of rice (*Oryza sativa* subsp. *japonica*) wuyujing3 in glass incubators at 24 °C under 16 h light/8 h dark photoperiod. RSV-carrying rates of viruliferous SBPHs

were screened by using dot enzyme-linked immunosorbent assay with the homemade monoclonal anti-NP antibody every three months (Zhao et al, 2016b).

*O. sativa* subsp. *japonica* cv. Nipponbare used for generation of knockout or overexpression strains was obtained by BioRun Technologies Co Ltd (Wuhan, Hubei, China). Wild-type (WT) and transgenic plants were grown in a greenhouse at 28 °C, 70% humidity under 12 h light/12 h dark photoperiod.

## RNA extraction, cDNA synthesis, and qPCR

Trizol reagent (Invitrogen, Carlsbad, CA, USA; 15596026) was used for total RNA extraction from salivary glands, guts, the entire bodies of SBPHs, or rice protoplasts, or rice leaves following the manufacturer's protocol. RNA quality was detected by a NanoDrop One (Thermo Scientific, Waltham, MA; 840-317400) and gel electrophoresis. For mRNA or viral RNA qPCR, 2 μg of RNA was reverse transcribed to cDNA by using the Superscript III First-Strand Synthesis System (Thermo Scientific; 18080051) and random primers (Promega, Madison, WI; C1181) (Zhao et al, 2016a). For miRNA qPCR, 2 μg of RNA was reverse transcribed by using miRNA RT enzyme mix of a miRcute plus miRNA first-strand cDNA kit (Tiangen, Beijing, China; KR211) (Zhao et al, 2021). qPCR amplification was performed on a LightCycler® 480 instrument II (Roche, Basel, Switzerland) using a LightCycler® 480 SYBR Green I Master (Roche; 04887352001), or a miRcute miRNA qPCR Detection Kit (Tiangen; FP411), respectively. In SBPH, *EF2* (Contig0.299) was used as an endogenous control for insect mRNAs and viral RNAs, and U6 snRNA was used as an endogenous control for miR-263a (Zhao et al, 2021). In rice, *UBQ10* (LOC4328390) or U6 snRNA served as an endogenous control for mRNAs and viral *NP*, or miR-263a. All qPCR amplifications are sequenced to verify the specificity of the primers. Six to eight figure replicates were prepared. Primers are listed in Appendix Table S1.

## Dual luciferase assays in *Drosophila* S2 cells

Dual luciferase assays were performed to assess the direct interaction of miR-263a and its target sites within SBPHs, following previously established protocol (Zhao et al, 2021). The 5′ untranslated regions (UTRs) of SBPH *CathBL* (PB.8570.2: 1 bp to 173 bp), *SPS5* (PB.9734.1: 881 bp to 1100 bp), and 3′ UTR of SBPH *NTFYbL* (PB.4359.9: 1542 bp to 1714 bp) containing the predicted miR-263a target sites were PCR amplified and cloned into the *Xho*I and *Not*I restriction sites of psiCHECK2 vector (Promega; C8021). For transfection, *Drosophila* S2 cells seeded in 24-well plates for 16 h were cotransfected with 1 μg recombinant psiCHECK2 plasmids and miR-263a mimics/NC (GenePharma, Jiangsu, China) at various concentrations (10 nM and 50 nM) using Lipofectamine 3000 (Invitrogen; L3000001). In addition, site-directed mutations in the complementary sequences to the "seed" sites of miR-263a were performed by a KOD-Plus mutagenesis kit (Toyobo, Osaka, Japan; F0936K) using specific primers. Co-transfection was then carried out with 1 μg of the mutated plasmids alongside 50 nM of miR-263a mimics or NC. miR-263a mimics are double-strand oligonucleotides corresponding to the sequence of mature miR-263a by GenePharma, with sequence 5′-AAUGGCAC UGGAAGAAUUCACGGG-3′ (sense) and 5′-CGUGAAUUCUUCC AGUGCCAUUUU-3′ (antisense). Cells were collected 16 h after

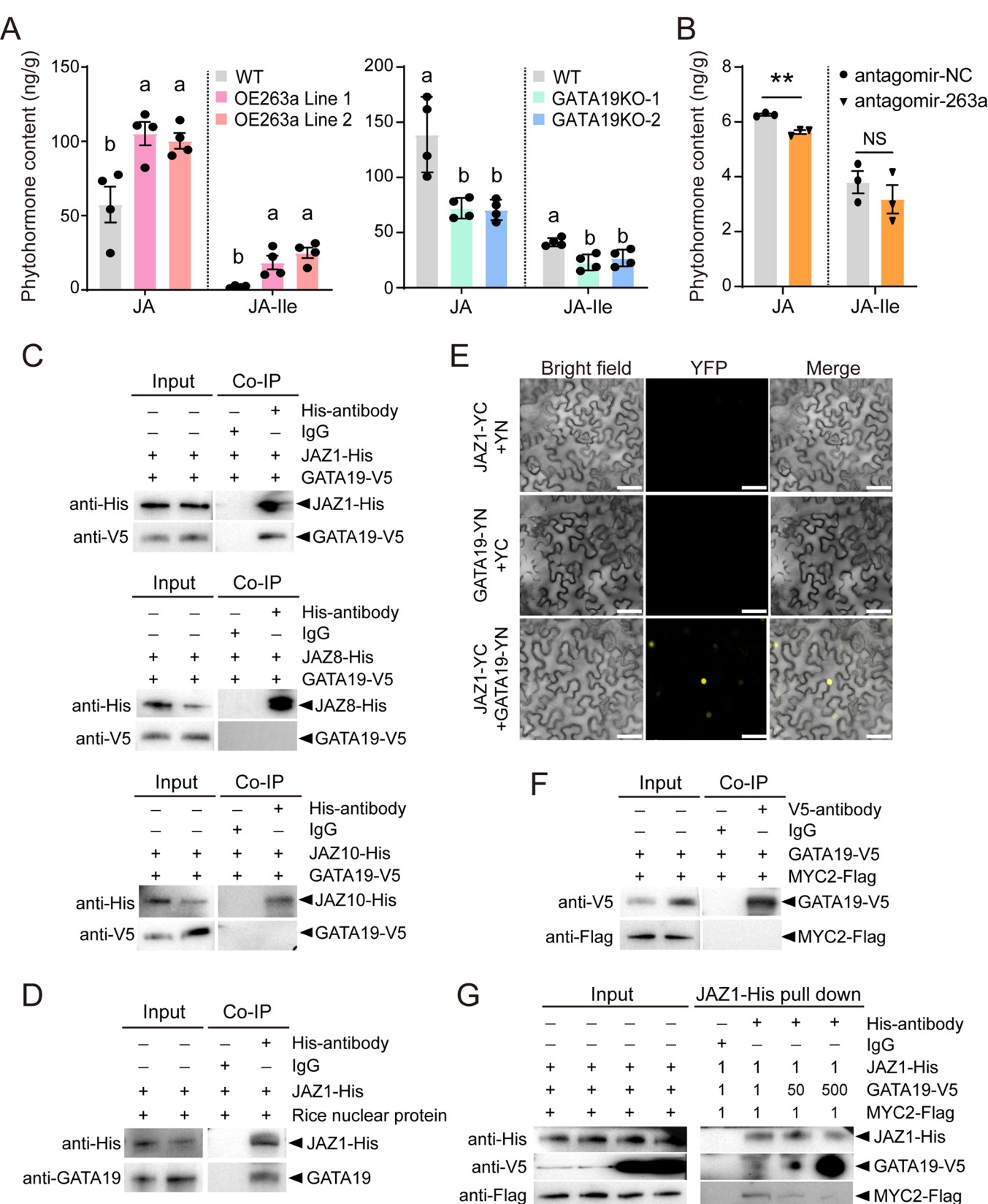

◀ **Figure 6.  GATA19 positively regulates jasmonate signaling pathway.**

(A) Quantification of JA and JA-Ile in WT, OE263a, and *GATA19*KO lines. Four biological replicates were prepared. Data are shown as mean ± SE. Comparisons among multiple groups were conducted using one-way analysis of variance (ANOVA) followed by Tukey's or Duncan's test. Different letters indicate significant differences. (B) Quantification of JA and JA-Ile in WT rice fed upon by nonviruliferous SBPHs with injection of antagomir-263a or NC. Three biological replicates were prepared. $P = 0.0015$. Values are shown as mean ± SE and were compared by Student's t test. NS, no significant difference. **$P < 0.01$. (C) Recombinantly expressed JAZ1/JAZ8/ JAZ10-His binds the recombinantly expressed GATA19-V5 in the Co-IP and Western blot assay. Mouse IgG was used as a negative control. (D) Recombinantly expressed JAZ1-His binds the GATA19 from rice nuclear extracts in the Co-IP and Western blot assay. (E) BiFC assay for the interaction of JAZ1-YC with GATA19-YN in *Nicotiana benthamiana* epidermal cells. Images were captured at 36 h post-infiltration. Scale bars: 20 μm. (F) Recombinantly expressed GATA19-V5 binds the recombinantly expressed MYC2-Flag in the Co-IP and Western blot assay. (G) His-tag pull-down assay for the competitive binding of JAZ1-His by GATA19-V5 and MYC2-Flag using recombinantly expressed proteins. The GATA19-V5 recombinant protein was incubated with JAZ1-His at varying concentrations, specifically at ratios of 1:50:500. Source data are available online for this figure.

transfection. The relative activity of Rluc normalized to Fluc activity is measured by a Dual-Luciferase Reporter Assay System (Promage; E1980) and is presented as the mean ± SE. Six replicates were prepared and assayed for each treatment. Primers used in this experiment are listed in Appendix Table S1.

## Target prediction of miR-263a in planthopper and in rice

miRanda (John et al, 2004) and RNAhybrid (Rehmsmeier et al, 2004) algorithms were used to identify the potential targets of miR-263a within SBPH transcripts (Tong et al, 2022), as described previously (Zhao et al, 2021). The primary principle for target prediction is "seed rules" which needs base paring from 2 to 8 nts of the mature miRNA. In addition, the minimum free energy (MFE) of the RNA duplex was analyzed by miRanda using a cutoff of −18 kcal mol$^{-1}$. The parameters of RNAhybrid were set as 1 hit per target, free energy threshold of −20 kcal mol$^{-1}$, helix constraint from 2 to 8, max bulge loop length of 2, max internal loop length of 8.

In rice, putative targets for miR-263a were predicted by psRNATarget (http://plantgrn.noble.org/psRNATarget/) (Dai and Zhao, 2011) to identify the potential targets (MSU Rice Genome Annotation, version 7), following parameters as below: number of top targets = 200, Expectation = 5, Penalty for G:U pair = 0.5, Penalty for other mismatches = 1, Extra weight in seed region = 0.5, Seed region = 2–14 nucleotides, Mismatches allowed in seed region = 0, HSP size = 19.

## Injection with miR-263a agomir or antagomir

Agomir of miR-263a was chemically modified dsRNA corresponding to the sequence of mature miR-263a by GenePharma (Zhao et al, 2021). The negative control (NC) sequence for miR-263a agomir was 5′-UUCUCCGAACGUGUCACGUTT-3′. A random sequence of 5′-CAGUACUUUUGUGUAGUACAA-3′ was synthesized as the NC sequence for the miR-263a antagomir. A total volume of 13.8 nL of agomir or antagomir or NC (200 μM; GenePharma) was delivered into the fourth-instar SBPH larvae by microinjection using a Nanoliter 2000 system (World Precision Instruments, Sarasota, Florida, USA). Four days after the injection, SBPHs were harvested for RNA or protein isolation.

## siRNA knockdown

dsRNA was generated using a T7 RiboMAX express RNAi system (Promega; P1700) and purified using a Wizard SV gel and PCR

clean-up system (Promega; A9282) following the manufacturers' protocols. A 489 bp of dsRNA for *Cathepsin B-like proteinase* (*CathBL*) (32–520 bp) and a 420 bp of ds*GFP* (73 bp to 492 bp) (23 nL of 6 mg mL$^{-1}$ solution) were delivered into the hemolymph of viruliferous fourth-instar larvae by microinjection through a glass needle using a Nanoliter 2000 system (World Precision Instruments).

## Inhibition of Cathepsin B protease

A 10 mM solution of CA-074 Me (ChemeGen, Los Angeles, California, USA; 147859-80-1), a membrane-permeable methyl ester form of CA-074, was prepared in 10% dimethyl sulfoxide (DMSO) (Sigma-Aldrich, Saint Louis, MO, USA; D2650). Viruliferous SBPH larvae were injected with 5 μM CA-074 Me or 1% DMSO. After 4 days, SBPHs were collected for qPCR and Western blot. Gut tissues were dissected for TUNEL assay and caspase assay, respectively.

## Detection of Caspases activity in vivo and in vitro

Guts from the viruliferous SBPH larvae injected with ds*GFP*/ ds*CathBL* (23 nL of 6 mg mL$^{-1}$ solution) for 4 d, or from viruliferous SBPHs after 4 d of treatment with DMSO/CA-074 Me, were collected for analyzing caspase activities using Caspase Assay Kit (orange, VAD-FMK) (Abcam, Cambridge, UK; ab270770), following the manufacturers' protocols. Nuclei were labeled with Hoechst (blue), and fluorescence was examined under a Zeiss LSM710 laser confocal microscope (Carl Zeiss AG, Oberkochen, Germany). Cells bearing active caspase enzymes covalently coupled to SR-VAD-FMK appear orange.

In addition, Caspase activities were also determined in the viruliferous SBPH larvae injected with ds*GFP*/ds*CathBL* for 4 d using human Caspase 3 and 8 Activity Assay Kit (Beyotime, Shanghai, China; C1116; C1151) and a SpectraMax Paradigm Multi-Mode Detection Platform (Molecular Devices, San Jose, CA, USA) according to the manufacturer's instructions. The standard curves were generated as described previously (Zhao et al, 2022b).

## Detection of Cathepsin B activity in vivo

In brief, guts from the viruliferous SBPH larvae injected with ds*GFP*/ds*CathBL* (23 nL of 6 mg mL$^{-1}$ solution) for 4 d were collected for analyzing Cathepsin B activity via the Zeiss LSM710 laser confocal microscope (Carl Zeiss AG) by using Cathepsin B Assay Kit (Magic Red) (Abcam; Ab270772) following the

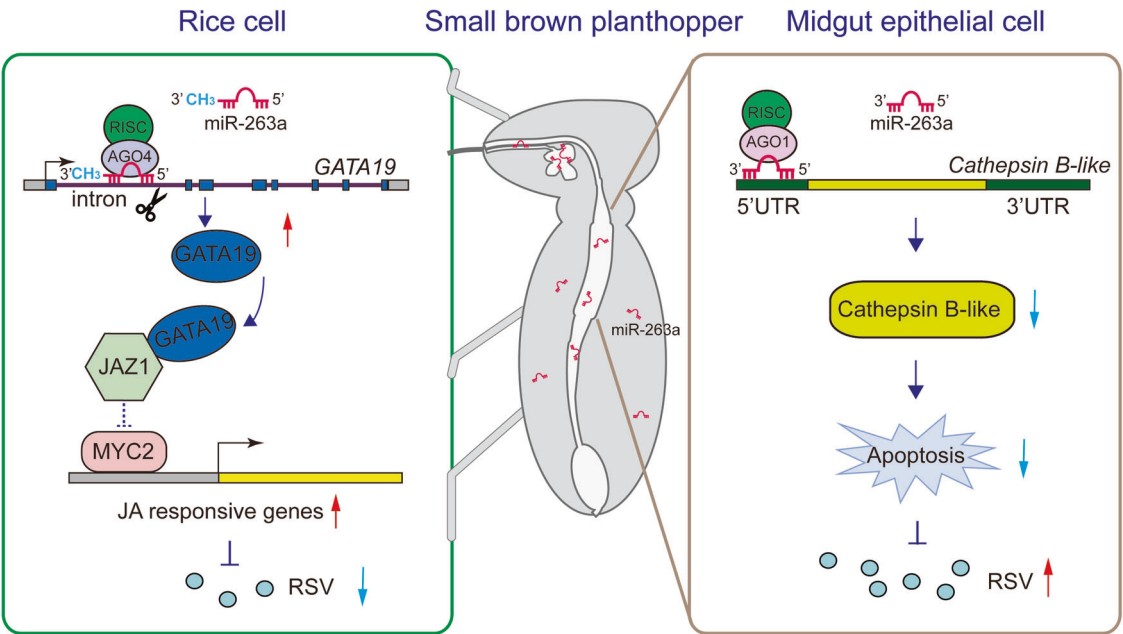

**Figure 7. A model illustrating the dual role of miR-263a in governing the lifecycle of RSV in both its insect vector and rice host.**

In SBPH midgut epithelial cells, miR-263a facilitates RSV accumulation through targeting a *Cathepsin B-like* gene to inhibit apoptosis. Upon transmitted to rice, miR-263a is methylated and cleaves the first intronic region of the transcription factor *GATA19*, leading to its heightened expression. Upregulation of GATA19 interferes with the interaction between JAZ1 and MYC2, thus activating jasmonate signaling pathway to afford rice resistance to RSV.

manufacturers' protocols. Nuclei were labeled with Hoechst (blue), and cells bearing active Cathepsin B activity appear red.

### Determination of the duration of miR-263a in rice

Ten nonviruliferous or viruliferous fourth-instar SBPH larvae were placed on each 3-week-old healthy rice seedling for 2 days. Subsequently, the SBPHs were removed, and rice leaves were collected immediately or at 6 or 12 days after SBPH feeding. Two leaves were collected as one replicate. Eight to ten replicates were set.

### Transient regulation of miR-263a amounts in rice leaves by SBPHs

To transiently promote miR-263a amounts in rice, viruliferous fourth-instar SBPH larvae were injected with agomir-263a (200 μM; 13.8 nL) for 4 d and confined in microcages on healthy rice leaves for a week. Rice fed on by SBPHs injected with agomir-NC served as the negative control. Inoculated rice leaves were then collected for miR-263a detection. Each rice leaf was inoculated with 10 insects confined in a microcage, and two leaves were collected per replicate. Eight to ten replicates were set.

Nonviruliferous fourth-instar larvae were injected with antagomir-263a or antagomir-NC (200 μM; 13.8 nL) for 2 d and then confined in microcages on healthy rice leaves for 2 d. After removing the insects, the rice seedlings were cultured for 1 d and then collected for *GATA19* transcript and JA quantification. Each rice leaf was inoculated with 10 insects. Two leaves per replicate and 10 to 12 replicates were collected for qPCR analysis. Two whole plants per replicate and 3 or 4 replicates were used for determining the JA contents.

### Extraction of rice nuclear and cytoplasmic RNAs

Rice nuclear and cytoplasmic proteins were extracted from viruliferous SBPHs-inoculated rice leaves by using a Plant nucleus and cytoplasm extraction kit (BestBio, Shanghai, China; BB-31124) according to the manufacturer's instructions. The reference proteins for nuclear and cytoplasmic proteins were histone H3 and Actin, which were detected using monoclonal anti-H3 antibody (EASYBIO, Beijing, China; BE3021) and anti-plant-Actin mouse monoclonal antibody (EASYBIO; BE0028), respectively. For each sample, four leaves totaling 150 mg were collected. RNAs were individually extracted from nuclear and cytoplasmic fractions by using Trizol reagent.

### RNA immunoprecipitation (RIP) assay

RIP assay was performed using a Magna RIP RNA-binding protein immunoprecipitation kit (Merck Millpore, Billerica, MA, USA; 17-700) as we previously described (Zhao et al, 2019a). Briefly, for miR-263a target validation, four days post injection with agomir-263a/NC, 20 fourth-instar viruliferous SBPH larvae were homogenized in ice-cold RIP lysis buffer and was incubated with magnetic beads and 2 μg of a homemade anti-Ago1 monoclonal antibody (Zhao et al, 2019a) or normal mouse IgG (Abcam; ab190475). One-tenth of the lysate supernatant served as the "Input" sample for reference. qPCR analysis was appropriately conducted to evaluate the fold enrichment of target RNAs in the immunoprecipitated (IP) fraction relative to the Input. The relative RNA level of each target segment, normalized to the IgG control, is presented as mean ± SE.

To detect which Ago miR-263a is loaded in rice, rice nuclear and cytoplasmic proteins were incubated with the beads and 2 μg of anti-

Ago1 or anti-Ago4 polyclonal antibodies (PhytoAB, San Francisco Bay Area, California, USA; PHY1381S, PHY0202S) or normal rabbit IgG (Abcam; ab171870). qPCR was performed to quantify the enrichment of target RNAs in the IP fraction compared to the Input, and the relative RNA level is presented as mean ± SE.

## Construction of miR-263a overexpression vector in rice protoplasts

A 450 bp sequence containing the osa-miR1876 precursor which could generate a 24-nt mature miRNA in rice (Wu et al, 2010), was amplified using KOD FX DNA polymerase (Toyobo; F0935K) with primers from Appendix Table S1. The obtained DNA fragments were inserted into the BamHI and SmaI sites of the pBI221 vector (MiaoLing Bio, Wuhan, China, P6700). Subsequently, the original miR-1876 and miR-1876* sequences in the recombinant pBI221 were replaced with miR-263a and miR-263a* sequences, creating pBI-OE263a plasmids obtained from GENEWIZ, lnc (Suzhou, China).

## Rice protoplast generation and transfection

Rice protoplasts were isolated from young rice seedlings aged 10–14 days according to the manufacturer's instructions (Coolaber, Beijing, China, PPT111-5T). Protoplasts with the viability up to 90% were transfected with 10 μg recombinant plasmids or along with 10 μL of RSV extracts obtained from RSV-infected rice leaves (Zhao et al, 2016b) using a PEG-mediated method for transient expression. After 36 h or 48 h of dark incubation, protoplasts were collected for qPCR and Western blot.

## miRNA target-EGFP reporter expression assay

miRNA target-EGFP reporter expression assay was used to assess the direct interaction of miR-263a and its target sites within plants. The 24-bp target sequence of miR-263a within the first intron of GATA19 was inserted just between the entire encoding region of EGFP and the terminator the pBI221-EGFP vector (HonorGene, Changsha, Huanan, China; HG-VZC0336), by using the In-Fusion HD Cloning kit (Takara, Tokyo, Japan; 638943). The recombinant pBI221-EGFP-target plasmids were transfected into rice protoplasts long with pBI-OE263a plasmids or the pBI221 empty vectors. Fluorescence was observed at 36 h post-transfection by using was under a Zeiss LSM710 laser confocal microscope (Carl Zeiss AG).

## Generation of transgenic rice plants

GATA19 knockout (GATA19KO) and miR-263a overexpression (OE263a) rice seedlings were obtained from BioRun Technologies Co., Ltd. (Wuhan, Hubei, China). KO rice lines were generated using CRISPR/Cas9-mediated mutagenesis. Two sgRNAs (5′-GGA GGGGTCTGACATGCCTCC-3′ for sgRNA1; 5′-AGCTGACGCTCG TGTACCAGGGG-3′ for sgRNA1) targeting the encoding region of the TIFY domain were designed and sequentially ligated into the binary vector binary vector pYLCRISPR/Cas9Pμbi-H. For construction of OE263a rice lines, the miR-263a precursor sequence were cloned from the recombinant pBI-OE263a plasmids, and were inserted into pBWA(V)HS vectors to generate pBWA(V)HS-ovmiR263 plasmids.

The recombinant plasmids were transformed into Nipponbare by the Agrobacterium-mediated method. GATA19 and OE263a lines were validated by sequencing with specific primers (Appendix Table S1). T2 homozygous lines were used to determine plant height, 1000-grain weight, husked grain width and length, and viral infection experiments.

## Oxidation and β elimination

RNAs extracted from rice leaves fed on by viruliferous SBPHs, viruliferous SBPHs, or in vitro synthesized miR-263a-OH were treated with NaIO4-mediated oxidation followed by β-elimination reaction as described previously (Alefelder et al, 1998). 5 ng of miR-263a-OH was used, which corresponded approximately to the amount of miR-263a from 100 mg of rice leaf tissues that inoculated with SBPHs. After precipitated using ethanol, miR-263a were then detected by Northern blot.

## Northern blot

Northern blot was performed to detect miR-263a in both insects and plants as described previously (Zhao et al, 2021). Two to three replicates were established, with each replicate containing either 30 insects or 100 mg of rice leaves. Briefly, RNA (10–15 μg) was separated by 15% (wt/vol) denaturing polyacrylamide gels and electroblotted onto nylon membranes (Invitrogen; AM10102). Biotin-labeled LNA oligonucleotide probes (GenePharma, Shanghai, China) were generated for the antisense sequences of mature miR-263a, LsU6 snRNA, or OsU6 snRNA. Probe sequences are listed in Appendix Table S1. Hybridizations with probes were conducted at 42 °C for miR-263a, while for LsU6 and OsU6, the temperature was set at 37 °C. After incubated with Streptavidin-HRP (Biodragon, Suzhou, Jiangsu, China; BF06185X), the chemical signals were detected by using SuperSignal West Femto (Thermo Scientific; 34094).

To assess CathBL mRNA decay, a 1134 bp fragment encompassing the 5′ UTR and coding region of CathBL was cloned into the KpnI and XhoI restriction sites of the pAC5.1B vector (Invitrogen; V411020). Drosophila S2 cells were cotransfected with 1 μg of the recombinant pAC5.1B plasmid and 50 nM of miR-263a mimic or NC (GenePharma). After 16 h of culture, the cells were collected for RNA extraction. RNA samples were electrophoresed through a 0.9% denaturing agarose gel and transferred to a positively charged nylon membrane (Invitrogen; AM10102) in 20×SSC buffer (Invitrogen; 15557044). A 189 bp probe for CathBL and a 223 bp probe for Drosophila Actin (NCBI accession number NM_078901.3) were synthesized using the Biotin Random Prime DNA Labeling Kit (Beyotime; D3118). Hybridization was carried out overnight at 42 °C in Skygen Nucleic Acid Hybridization Solution (Tiangsa, Beijing, China; 220566) with probes at 20 ng/mL. After incubation with Streptavidin-HRP (Biodragon; BF06185X), signals were detected using SuperSignal West Femto (Thermo Scientific; 34094). Primers are listed in Appendix Table S1.

## Phytohormone quantitative assay

Approximately 150 mg of stems and leaves from 4-week-old WT, OE263a, and GATA19KO rice lines, as well as rice fed on by nonviruliferous SBPHs with injection of miR-263a antagomir or

control antagomir, were finely ground into powder. Measurement of JA and JA-Ile was conducted using high-performance liquid chromatography-mass spectrometry (LCMS-8040; Shimadzu Corporation, Kyoto, Japan) following established protocols (Lu et al, 2020). $D_6$-JA and $^{13}C_2$-JA-Ile served as internal standards. Concentration was calculated in nanograms of phytohormone in each gram of plant fresh weight, with results reported as the mean ± SE ($n = 3$ or 4).

## 5′ RLM-RACE of rice mRNA cleavage products

In order to detect the cleavage site of miR-263a in rice, 5′ RLM-RACE was performed by using FirstChoice RLM-RACE (Thermo Scientific; AM1700M) following the manufacturer's instructions. Briefly, total RNA was extracted from OE263a lines and directly ligated to the 5′ RACE Adapter without further modification. Random decamers were used to prime cDNA synthesis with reverse transcriptase. 5′ RACE gene specific outer and inner primers are listed in Appendix Table S1. All 5′ RLM-RACE PCR products were analyzed on agarose gels, purified, cloned, and sequenced.

## Validation of miR-263a promoting intron splicing in vitro

This assay was performed in rice protoplasts. Initially, a 2596 bp sequence was obtained, spanning from 1 to 2593 bp of the *GATA19* DNA sequence following the exon-intron-exon reading direction and included a stop codon TAA. Subsequently, this 2596 bp sequence replaced the GUS encoding region within the pBI221 vectors, resulting in the creation of pBI221-splicing plasmids (*Ex-In-Ex*) by using the In-Fusion HD Cloning kit (Takara; 638943). In addition, site-directed mutations in the 17-nt binding sequence of miR-263a within the intron region were performed by a KOD-Plus mutagenesis kit (Toyobo; F0936K) using specific primers, to construct *Ex-InMT-Ex* plasmids. These recombinant plasmids, along with either pBI-OE263a or the pBI221 vectors, were transfected into rice protoplasts. Protoplasts were then harvested 36 h post transfection. Quantification of the transcript of *GATA19* was performed using qPCR with primers Ex*GATA19*-F/R. Primers are listed in Appendix Table S1.

## Protein expression and Co-IP assay

Rice proteins including JAZ1, JAZ10, GATA19, JAZ8, and MYC2 were synthesized using Premium ONE Expression Kit (Tiangen; CFS-EDX-ONE) following the manufacture's protocol. The respective ORFs were cloned into the pEU-E01-MCS-TEV-His vector at designated restriction sites: JAZ1, JAZ10, and GATA19 at XhoI/BamHI; JAZ8 at SpeI/BamHI; and MYC2 at EcoRV/SpeI. For expression, 2.5 μL of each recombinant plasmid (15 ng μL$^{-1}$) was mixed with 2.5 μL WEPRO TTmix, layered onto the bottom of a 0.2 mL PCR tubes filled with 50 μL SUB AMIX TT to form a double layer, and incubated at 26 °C for 24 h.

For Co-IP analysis, 5 μg of anti-His monoclonal antibody (EASY-BIO; BE2019) or anti-V5 monoclonal antibody (EASYBIO; BE2032) was initially incubated with 20 μL of Dynabeads Protein G (Invitrogen; 10003D) as previously described (Lu et al, 2020). The mixture was then exposed to either a mixture of recombinant JAZ1-His and nuclear proteins extracted from healthy rice, or a 1:1 mixture of recombinant JAZ1-His/JAZ8-His/JAZ10-His and GATA19-V5, or a 1:1

combination of GATA19-V5 with MYC2-Flag, and allowed to interact for 2 h at 4 °C. In the competitive interaction assay, we manipulated the concentration of GATA19-V5 recombinant protein ratios of 1:50:500. Following the 1-h incubation of GATA19-V5 with JAZ1-His, MYC2-Flag was introduced and incubated for an additional hour. Mouse IgG (Abcam; ab6708) served as the negative control. Post incubation, the complexes were washed with washing buffer before being released from the beads with elution buffer for subsequent Western blot analysis.

## Western blot

Proteins extracted from SBPHs, rice protoplasts or transgenic lines, or rice proteins expressed in vitro were loaded onto the JESS automated Western Blot system (ProteinSimple) according to the manufacture, or separated by SDS-PAGE and detected by using SuperSignal West Femto (Thermo Scientific; 34094). For JESS, concentrations of anti-NP (Zhao et al, 2016b) and anti-GAPDH (Wang et al, 2013) (1:200) were adjusted to the technology. The NP signal was then quantified relative to the GAPDH loading control for each sample. Data analysis was performed using Compass software (Bio-Techne, TECH, US) (https://www.proteinsimple.com/compass/downloads).

For traditional Western blot, rice proteins were detected using Ago1 polyclonal antibodies (PhytoAB; PHY1381S), Ago4 polyclonal antibodies (PhytoAB; PHY0202S), anti-H3 (EASYBIO; BE3021), anti-plant-Actin (EASYBIO; BE0028), anti-V5 (EASYBIO; BE2032), anti-Flag (Abcam; ab205606), anti-His (EASYBIO; BE2019) monoclonal antibodies respectively. The immunoblot signal was detected using SuperSignal West Femto (Thermo Scientific; 34094).

## Bimolecular-fluorescence complementation (BIFC) assay

BiFC assay on the interaction between GATA19 and JAZ1 were performed as described previously (Ge et al, 2024). The ORFs of *GATA19* and *JAZ1* were cloned into binary nYFP or cYFP vectors using ligation-independent cloning. Primers are listed in Appendix Table S1. These two recombinant plasmids or each recombinant vector (negative control) were transformed into *Agrobacterium* GV3101 (Zoman, Beijing, China; ZK295) and infiltrated into *N. benthamiana* leaves. Two days post-infiltration, YFP fluorescence was visualized using a Zeiss LSM710 laser confocal microscope (Carl Zeiss AG) (Zhou et al, 2009).

## Rice disease incidence assay

Each two-week-old leave of the WT, OE263a, or *GATA19*KO rice plants was fed on by 2 viruliferous fourth-instar SBPH larvae that were trapped in a microcage for a duration of 7 days. Following insect removal, disease symptoms were observed in the rice leaves incubated in a greenhouse at 28 °C. Five plants per replicate and six replicates were used to calculate the disease incidences.

## Statistics

Unpaired two-tailed Student's t test was used to compare the means of the two groups after confirming that the data met the assumptions of normality and equal variances. To assess normal distribution, we applied the D'Agostino-Pearson normality test and

the Brown-Forsythe test for equal variances. For comparisons involving more than two groups, one-way analysis of variance (ANOVA) followed by Tukey's test was employed. In case where the data did not follow a normal distribution, we used the nonparametric Mann–Whitney U test for comparing two experimental groups and the Kruskal–Wallis one-way ANOVA followed by Dunn's post hoc test for analyzing multiple experimental groups. Statistical analysis was performed with GraphPad Prism software version 9.0 or SPSS statistics 19.0. Results are given as dot plots or histograms with mean ± SEM, as specified in the figure legends. A *P* value < 0.05 was considered statistically significant. Four to twelve replicates were prepared and analyzed for each experimental group. For qPCR, only samples with reference genes showing normal CT values were included for further analysis; otherwise, samples were excluded.

## Data availability

Gene and protein sequences have been deposited in the NCBI (GenBank number PP716112: https://www.ncbi.nlm.nih.gov/nuccore/PP716112). Software used in this study can be found in the Methods section.

The source data of this paper are collected in the following database record: biostudies:S-SCDT-10_1038-S44318-025-00405-4.

## Peer review information

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

## Acknowledgements

This work was supported by the grants from the National Key R&D Program of China (2022YFD1401700), the National Natural Science Foundation of China (No. 32230090, 32425047, 32272532), and the Youth Innovation Promotion Association of CAS (No. Y2023022).

## Author contributions

**Wan Zhao**: Conceptualization; Funding acquisition; Investigation; Methodology; Writing—original draft; Writing—review and editing. **Hong Lu**: Investigation. **Jiaming Zhu**: Investigation. **Lan Luo**: Visualization. **Feng Cui**: Conceptualization; Funding acquisition; Writing—review and editing.

Source data underlying figure panels in this paper may have individual authorship assigned. Where available, figure panel/source data authorship is listed in the following database record: biostudies:S-SCDT-10_1038-S44318-025-00405-4.

## Disclosure and competing interests statement

The authors declare no competing interests.

# Expanded View Figures

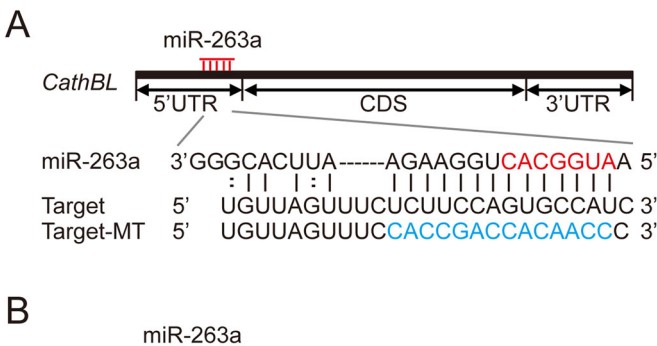

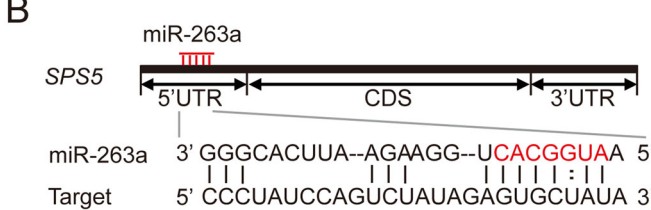

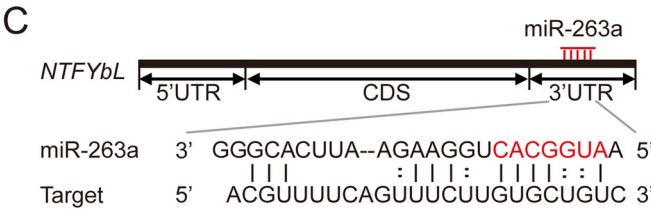

**Figure EV1. Sequence alignments of miR-263a with the candidate target genes of SBPH.**

(A) *Cathepsin B-like* (*CathBL*). The mutated target (Target-MT) is also shown. (B) *Nuclear transcription factor Y subunit beta-like* (*NTFYbL*). (C) *Serine protease snake-5* (*SPS5*). The miR-263a seed sequence is highlighted in red and the mutated sequences are indicated in blue.

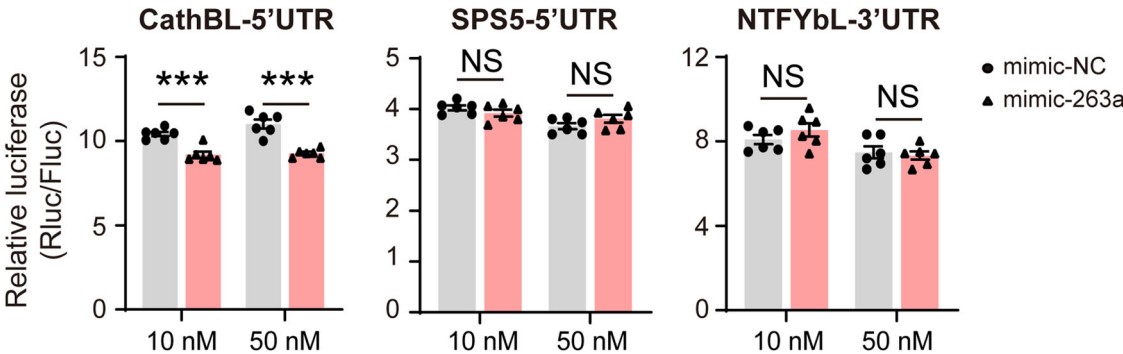

**Figure EV2. Luciferase reporter assays in *Drosophila* S2 cells cotransfected with miR-263a/NC mimics and psiCHECK2 vectors containing *CathBL* 5′UTR, *SPS5* 5′UTR, and *NTFYbL* 3′UTR as the candidate targets of miR-263a.**

*P* values from left to right, $P = 2.83E-04$, $P = 1.27E-04$. Six biological replicates were prepared. Values are shown as mean ± SE and were compared using Student's t test. NS, no significant difference. ***$P < 0.001$.

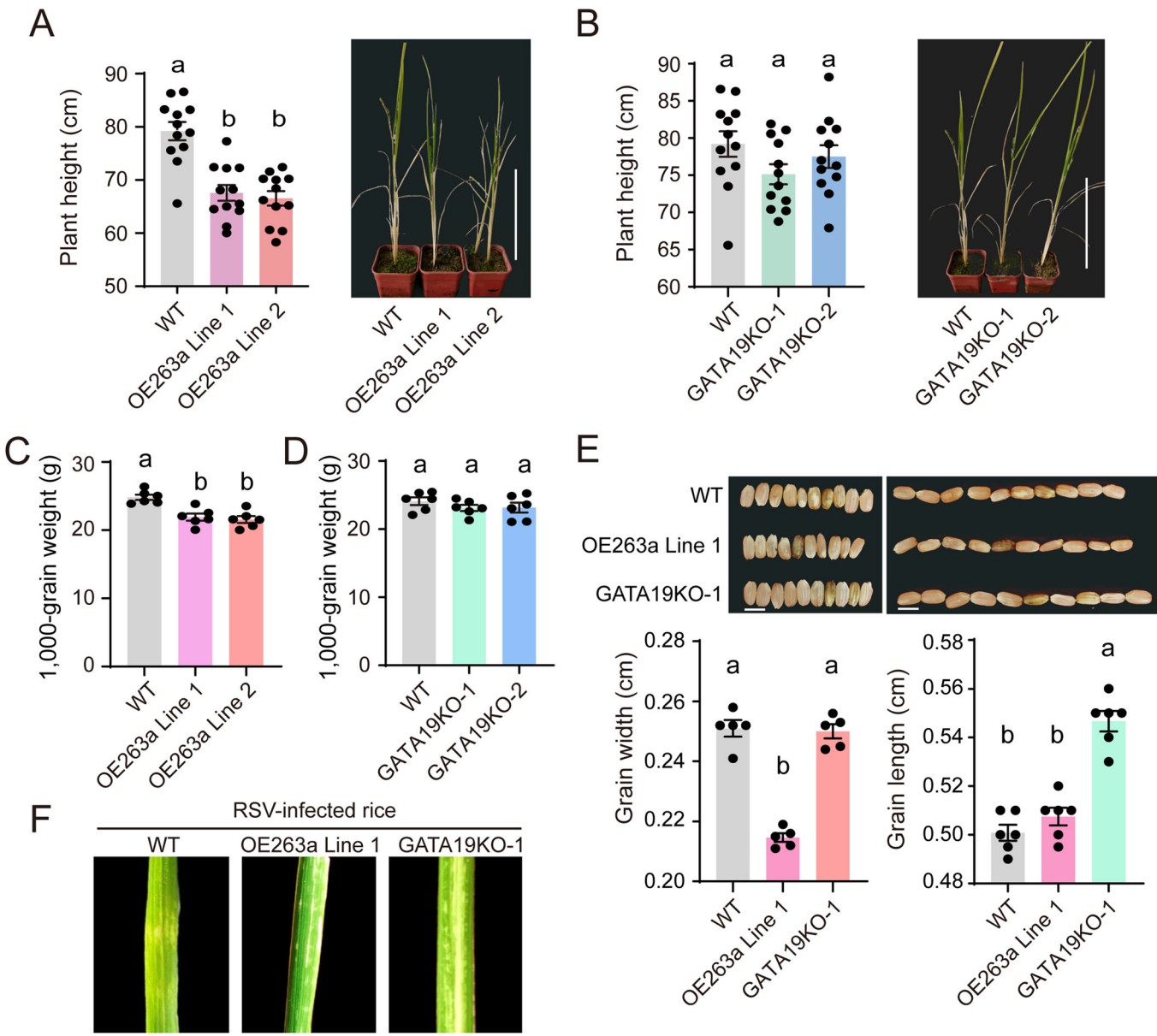

**Figure EV3.  Gross morphology of rice plants.**

(**A**) Comparative plant height between WT and OE263a lines, with a scale bar representing 50 cm. Twelve biological replicates were prepared, with each replicate including one rice plant. (**B**) Plant height comparison between WT and *GATA19*KO lines, utilizing the same scale bar for reference. Twelve biological replicates were prepared, with each replicate including one rice plant. (**C**) The 1000-grain weight of WT and OE263a lines. Six biological replicates were prepared. (**D**) The 1000-grain weight of WT and *GATA19*KO lines. Six biological replicates were prepared. (**E**) Husked grain width and length of WT, OE263a, and *GATA19*KO lines. Scale bars: 5 mm. Five to six biological replicates were prepared, with each replicate including 10 grains. (**F**) Disease symptom of WT, OE263a, and *GATA19*KO lines after inoculation with RSV. For (**A**) to (**E**), the values are reported as mean ± SE. Data comparisons among multiple groups were performed using one-way analysis of variance (ANOVA) followed by Tukey's test. Different letters indicate significant differences.

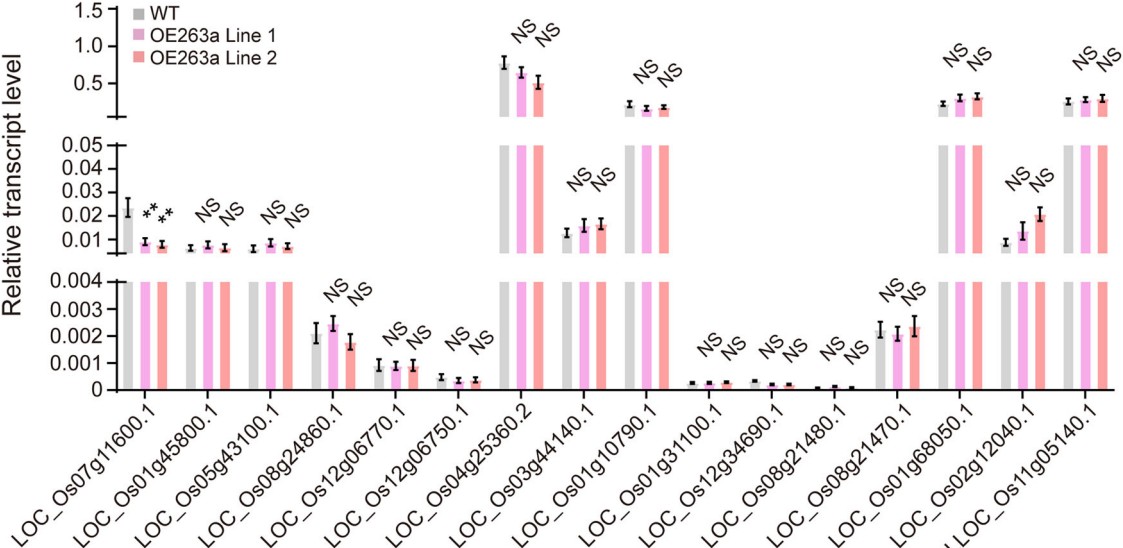

**Figure EV4.** Relative transcript levels of 16 candidate target genes of miR-263a in WT and OE263a rice lines.

Eight biological replicates were prepared, with each replicate including two leaves. $P$ values from left to right, $P = 0.0041$, $P = 0.0062$. Values are reported as mean ± SE and were compared by Student's t test. NS, no significant difference. **$P < 0.01$.

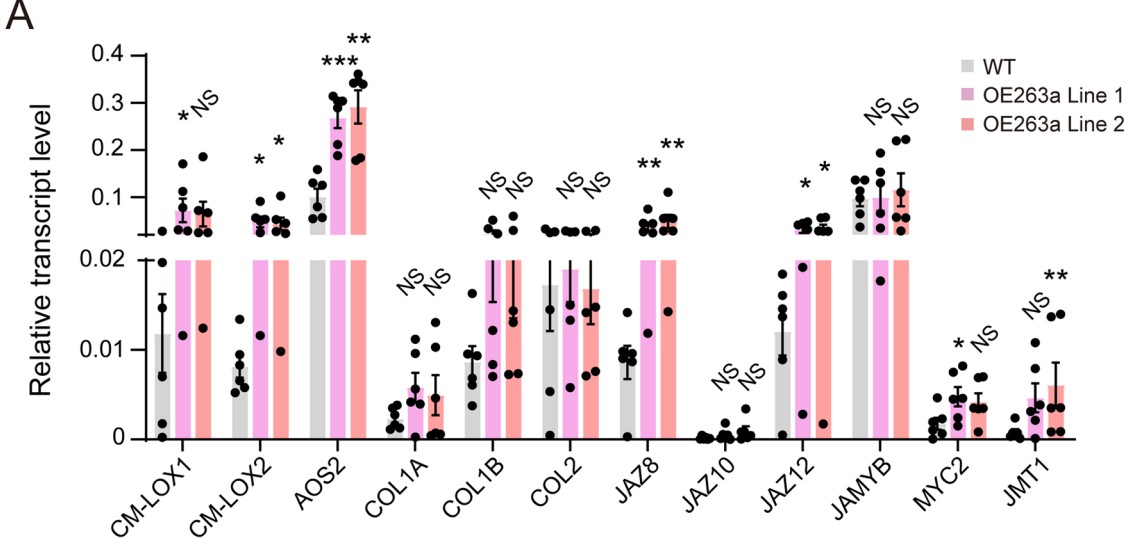

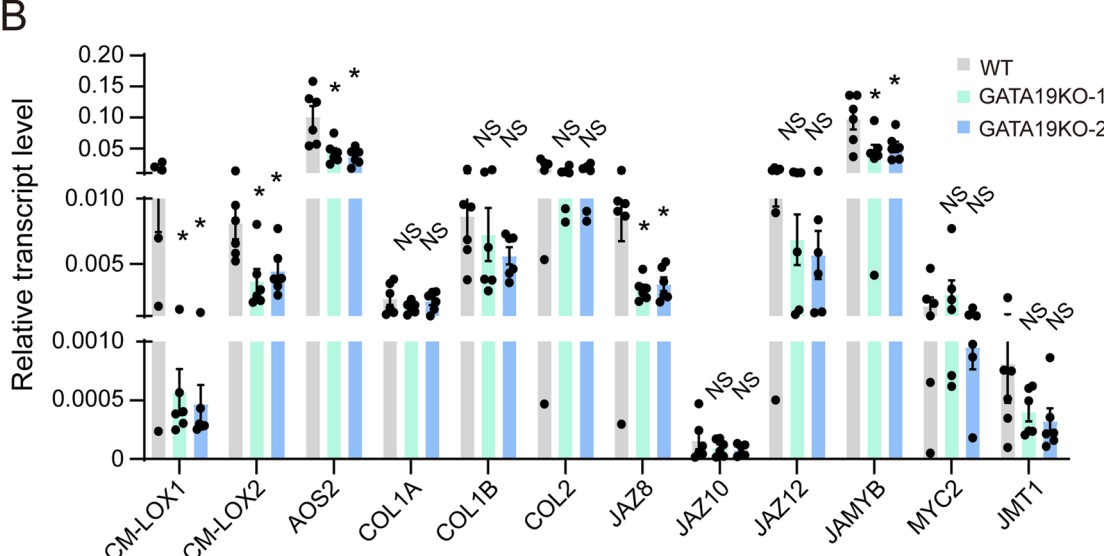

**Figure EV5. Relative transcript levels of 12 genes relative to jasmonate signaling pathway in WT and OE263a lines (A), or in WT and *GATA19*KO lines (B).**

Six biological replicates were prepared, with each replicate including two leaves. For (**A**), *P* values from left to right, $P = 0.0395$ (*CM-LOX1*), $P = 0.0171$ (*CM-LOX2*), $P = 0.0476$ (*CM-LOX2*), $P = 1.45E{-}04$ (*AOS2*), $P = 0.0022$ (*AOS2*), $P = 0.0043$ (*JAZ8*), $P = 0.0022$ (*JAZ8*), $P = 0.0299$ (*JAZ12*), $P = 0.0398$ (*JAZ12*), $P = 0.0400$ (*MYC2*), $P = 0.0087$ (*JMT1*). For (**B**), *P* values from left to right, $P = 0.0260$ (*CM-LOX1*), $P = 0.0260$ (*CM-LOX1*), $P = 0.0152$ (*CM-LOX2*), $P = 0.0296$ (*CM-LOX2*), $P = 0.0129$ (*AOS2*), $P = 0.0134$ (*AOS2*), $P = 0.0260$ (*JAZ8*), $P = 0.0260$ (*JAZ8*), $P = 0.0252$ (*JAMYB*), $P = 0.0354$ (*JAMYB*). Values are presented as mean ± SE and were compared by Student's t test. NS, no significant difference. *$P < 0.05$. **$P < 0.01$. ***$P < 0.001$

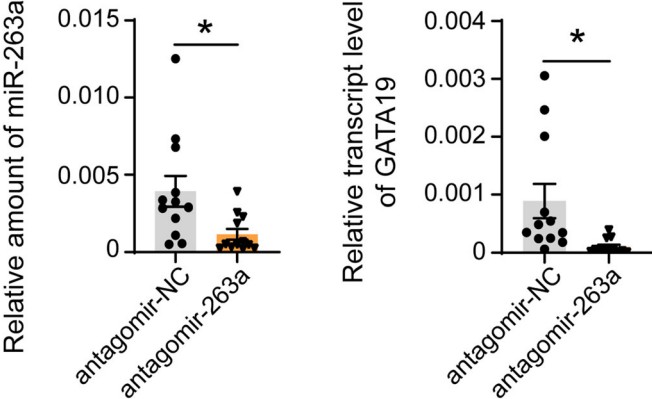

**Figure EV6.  Relative amount of miR-263a and relative transcript level of *GATA19* in wild-type rice fed upon by nonviruliferous SBPHs with injection of antagomir-263a or NC.**

OsU6 snRNA and *UBQ10* serve as internal references for miRNA and gene, respectively. Twelve biological replicates were prepared, with each replicate including two leaves. *P* values from left to right, *P* = 0.0202, *P* = 0.0211. Values are presented as mean ± SE and were compared by Student's t test. *$P < 0.05$.

