## [Peer Review File · The EMBO Journal]

A double-agent microRNA regulates viral cross-kingdom infection in animals and plants

Feng Cui, Wan Zhao, Hong Lu, Jiaming Zhu, and Lan Luo

Corresponding author(s): Feng Cui (cuiif@ioz.ac.cn) , Wan Zhao (zhaow@ioz.ac.cn)

Review Timeline:

Submission Date:	16th Aug 24
Editorial Decision:	10th Oct 24
Appeal Received:	6th Jan 25
Editorial Decision:	12th Feb 25
Revision Received:	14th Feb 25
Accepted:	24th Feb 25

Editor: Ieva Gailite

Transaction Report:

Dear Dr. Cui,

Thank you for submitting your manuscript for consideration by The EMBO Journal. We have now received a full set of reviewer reports, which are included below for your information. Based on these comments, we unfortunately had to conclude that the study is not a sufficiently strong candidate for publication in The EMBO Journal.

As you will see, while the reviewers per se express interest in the proposed role of a small brown planthopper miRNA in regulation of rice stripe virus infection both in the insect host and the rice plant. However, they also find that the proposed function of miR-263a in regulation of insect host infection and plant immune defense response is currently insufficiently substantiated. Since major experimental revisions with an uncertain outcome would be needed to address these referee concerns, I am afraid that we cannot extend an invitation to revise the manuscript.

Nevertheless, if you find that you can address all or most of the reviewers' concerns, I would be willing to reconsider the revised manuscript. In such a case, I would send it back to the same reviewers, if possible, but would allow them to make new comments on the data, which might then have to be further addressed if the reviewers are more positive in this round of assessment.

Thank you in any case for the opportunity to consider this manuscript. I regret that I could not communicate more positive news, but I nevertheless hope that you will find our reviewers' comments helpful for further improvement of the manuscript.

Yours sincerely,

Ieva Gailite

Referee #1:

The manuscript by Zhao et al. provides interesting observations that the dual-roles of miR-263a in regulating rice stripe virus infection in both insect vector and host plants. In insect vectors, miR-263a target the cathepsin B-like gene to inhibit apoptosis, which facilitates RSV replication. After delivery of insect vectors, miR-263a improves the mRNA level of GATA19 and then activates JA signaling pathway, which trigger antiviral immunity. Overall, the manuscript provides an interesting story of miR-263a in modulating viral cross-kingdom infection. However, some underlying mechanisms remain to be elucidated. Some conclusions and descriptions are confusing and need be clarified by more evidence.

Major comments:

1. It is very interesting to show the translocation of miR-263a from insect to plants. The major concern is whether the amount of insect-injected miR263a is enough to induce change of GATA19 mRNA and downstream JA signaling. The GATA19 mRNA accumulation and JA signal induction can be directly compared in the plants fed by healthy insects and insects with antagomir-263a injection.
2. In the Figure 2, it should be clarified whether the binding of miR263a to the 5'UTR of CathBL induces mRNA cleavage or translation inhibition.
3. In Figure 3, it needs to be clarified whether the pre-miR263a or mature miR-263a are injected into plant cells. In addition, in Figure 3B, C, and D, the method for analyzing accumulation of miR263a should be provided in the figure legend, from northern blotting or deep sequencing. In addition, what's the means of the relative values in the Y axis?
4. In Figure 5, all the figure legend and result description are confusing and cannot support the conclusion that miR-263a elevates GATA19 expression through into cleavage. For instance, in 5E, there are some inconsistencies between the figure legend and labels in the figure. Is the result accumulation of GATA19 or miR263a? It needs more evidence to show that the cleavage in the middle of intron could improve the mRNA accumulation.
5. In Figure 4G and 5G, the virus disease symptoms should be shown.

Minor Comments:

- 1: The disease incidence rates (only 0.4% or 0.8%) are too low, and it is something wrong with the unit of Y axis.

- 2: In figure 6C, the negative control of GATA19-YN and empty YC is lack.
- 3: In the proposed model, it is better to label the 5' and 3' end of miR263a.

Referee #2:

This manuscript presents a study about the function of one insect microRNA miR-263a on virus accumulation in insect vector and rice host. The miR-263a was considered to inhibit cathepsin B-mediated apoptosis to facilitate viral proliferation in midgut epithelial cells of insect. However, it has not been clearly explained why the expression of miR-263a in insect infected by virus significantly decreased. In rice, this exogenous miR-263a might activate jasmonate signaling pathway by upregulates the expression of transcription factor GATA19 to interfere JAZ1 binding with MYC2, which leading to rice antiviral activity. However, it is essential for the authors to provide more experimental evidence. In addition, the Figures should be displayed in order. The order of Figure 2, 3 and 5 is somewhat disorderly, especially Figure 5, the marks in the paper do not correspond to the Figure, and Figure 5E is not mentioned in the paper.

Authors found that miR-263a positively regulated the accumulation of viruses in insect vectors, but its expression decreased significantly in the intestines and salivary glands of viruliferous insects (Fig.1B). What was the expression level of miR-263a at different times when the virus infected insects? Is the expression of miR-263a inhibited after virus infection? In addition, the expression of miR-263a decreased significantly in the intestines and salivary glands of viruliferous insects, but there was no significant difference in the expression of miR-263a in the whole insects. Is the expression of miR-263a in other tissues high? The detection of disease resistance of both miR-263a overexpression plants and GATA19 knockout plants showed significant differences in disease resistance after 12 days (Fig. 4H, Fig.5H), but miR-263a began to decline 6 days after it was transferred from insects to plants, and almost disappeared after 12 days (Fig. 3C). How to explain these results?

miR-263a up-regulates the JA pathway, but down-regulates the SA pathway at the same time (Fig. 6C). If miR-263a negatively regulates the accumulation of viruses in rice from the SA pathway, these results need to be discussed

Major revision:

1. It can be seen from the Fig. 1A that rRNA of SBPH in line1 is degraded, this may have led to significant differences in pre-miR-263a and miR-263a between the two insect groups. Interestingly, the expression of miR-263a decreased in the gut and salivary glands of viruliferous insects, while the decrease was not significant in the whole body. It is recommended to use northern to verify the expression level. The NP-RNA level of RSV increased significantly with the up-regulation of miR-263a level. This result indicated that miR-263a was conducive to the accumulation of RSV in SBPH, but there was no direct evidence to prove that it was involved in viral replication.
2. Similarly in the results of line1 19 paragraph, there is no direct evidence that CathBL is involved in viral replication, which involves viral genome replication and assembly, not just changes in NP level.
3. In Fig. 2G and 2J, the two figures need to be merged to see the cells clearly and make sure that the label is in the cell layer.
4. In Fig. 2H, GFP-treated insects exhibited predominantly nuclear localization of TUNEL staining, whereas DMSO treatment resulted in non-exclusive nuclear localization of TUNEL. Is TUNEL located in the nucleus? The presence of NP is barely discernible in the cellular layer following DMSO treatment, whereas it is predominantly localized within the intestinal epidermis. There is also no significant difference in NP between DMSO and CA-074 methyl ester treatment.
5. In Fig. 3E, the internal reference LsU6 snRNA and OsU6 snRNA of some samples were not hybridized, please explain the reason. In addition, there was the mobility shift of miR-263a in rice samples, which was not consistent with the result description. The two bar charts in Fig. 3F need to be distinguished.
6. How is the difference of NP relative to actin calculated in Fig. 4B? It is obvious from the Westernblot that the ratio is greater than 1, which is inconsistent with the bar figure. These data need to be checked and calculated.
7. There is a messy ordering problem that makes it difficult to read and understand in Fig. 5. The marks in the manuscript do not correspond to the figure. There are A-H in the figure and A-G in the manuscript. The Fig. 5E is not mentioned in the manuscript, and the 5E mentioned in the manuscript is actually 5F in the Figure.

Referee #3:

In this ms, the authors extend their previous work of the identification of miR263a as a regulator of the RSBV infection in insects to further address its function cross kingdom both in the insect and plant host rice. Using agomir and antagomir approaches they increased or reduce miR263 accumulation and they identified potential targets both in the insect and the plant host. They confirmed using transient assays that these targets are cleaved in these systems suggesting that a similar regulation may occur in vivo. For the insect, they inject dsRNA against the CathBL target and linked this gene to cell death as it is well known for these regulators. Then, they found miR263 in insect saliva suggesting the secretion of this miRNA inside the plant host. Then, they found that the insect miRNA is methylated in planta and loaded into AGO4, a known actor for 24nt miRNA action in rice. They also create transgenic plants overexpressing miR263 or inhibiting their action as well as the previous approaches to dissect how miR263 can regulate infection and their targets. There is a huge amount of work and the results are interesting and maybe attractive to a broad audience. However, the paper is very difficult to read and also the legends to figures are not with the figures making more difficult to read as they are very complex. Furthermore, Figs 2 and 3 are mentioned at different times in the text and the logical of the organization is not easy to follow particularly for a non-expert reader in the miRNA field.

There are two major points that seems crucial to show that the post-transcriptional regulation of miR263 is relevant cross kingdom. Indeed, the fact that dsCath affects RSBV infection shows that the protein is required for this response but not that the miR263 post-transcriptional regulation is involved in the process. Since the miR263 is from insect origin its action on an insect mRNA, CathDBL, to use a miR-resistant version is less crucial. However, the same is particularly true for the post-transcriptional regulation of GATA19 by miR263a in a "cross-kingdom" manner. It is required to express miR-resistant versions of these transcripts in plants and test the phenotype in correlation with the increased stability of the GATA19 during infection. The biological relevance of miR263 action in plants is supported by expressing miR263 although it is crucial to show that a plant target is regulated by this cleavage.

In addition to this major point, several other points need to be addressed/

1. Fig 2B; Two genes are not detected and cannot be concluded whether they are regulated or not with these panels? Or this has to be shown using a different scale on the X?
2. Fig. 2B Why NTFYbL has a positive instead of a negative if this gene is a target? Can you try to explain? OR if not discard this "target" too.
3. Fig. 2C the differences are very minor at 10nm compare to Fig. 2D? Although I understand that the miR-resistant mutant experiment suggest that this difference is significant. This is the mutant version I previously mentioned that could be used to test the role of miR263 regulation in RSBV infection.
4. Fig. 3A is normalized by rRNA? There is large variation in U6 probes which are also constitutive. Are other miRNAs NOT transported to the host that can be used as controls for "transport".
5. Fig. 3B and 3C what do you mean by 0 levels in a qPCRs? There is no difference of miR263 levels between viliferous and non viliferous interactions? Any explanation?
6. Fig. 3D, this is the type of kinetics that needs to be done with plants expressing a miR-resistant version of the target to conclude about miR263 action on this target.
7. The experiments on protoplasts to show that GATA19 is upregulated by miR263Oex in transient assays and plants support the idea that miR263 regulates GATA19 and a version miR-resistant on the intron sequence could be used in the transient assay to test this regulation (even isoforms expression to see global impact on alternative splicing of this gene?). Clearly that a GATA19 KO regulates defense genes (jasmonate pathway) can be through many mechanisms (it is a TF!) and a GATA19 miR-resistant must reduce Jasmonate levels and clearly show that the targeting of miR263 for GATA 19 is relevant in vivo.

** As a service to authors, EMBO Press provides authors with the possibility to transfer a manuscript that one journal cannot offer to publish to another EMBO publication or the open access journal Life Science Alliance launched in partnership between EMBO Press, Rockefeller University Press and Cold Spring Harbor Laboratory Press. The full manuscript and if applicable, reviewers' reports, are automatically sent to the receiving journal to allow for fast handling and a prompt decision on your manuscript. For more details of this service, and to transfer your manuscript please click on Link Not Available. **

The comments are in blue and our responses are in black.

Referee #1:

Major comments:

1. It is very interesting to show the translocation of miR-263a from insect to plants. The major concern is whether the amount of insect-injected miR263a is enough to induce change of *GATA19* mRNA and downstream JA signaling. The *GATA19* mRNA accumulation and JA signal induction can be directly compared in the plants fed by healthy insects and insects with antagomir-263a injection.

Reponses: This is an excellent suggestion. We measured the change of *GATA19* transcript levels and JA and SA contents in rice plants fed by nonviruliferous SBPHs with antagomir-263a or antagomir-NC injection. After 2 d of injection, SBPHs were allowed to feed on rice seedlings for 2 d. The rice seedlings were cultured for an additional day before collected for further measurement. We found that both miR-263a and *GATA19* transcript levels were significantly reduced in rice plants fed upon by SBPHs with antagomir-263a injection compared to those with antagomir-NC injection. Correspondingly, the JA content decreased significantly, whereas JA-Ile and SA contents remained unchanged. These results suggest that insect-injected miR-263a primarily influences the JA signaling pathway in rice under natural conditions. Therefore, we focus on the effect of miR-263a on the JA signaling pathway. These new results are presented in the new Fig. 6B and Fig. EV6, Results part (Lines 298-305) and Materials and Methods part (Lines 529-535). The description of SA is deleted in new Fig. 6A and Results part.

2. In the Figure 2, it should be clarified whether the binding of miR263a to the 5'UTR of *CathBL* induces mRNA cleavage or translation inhibition.

Reponses: To clarify whether miR-263a induces the decay of *CathBL* mRNA, we performed a northern blot assay in S2 cells, in which *CathBL* mRNA containing the 5' UTR was expressed with co-transfection of miR-263a mimic or NC. A biotin-labeled probe for *CathBL* was applied. A significant decrease in the full-length *CathBL*

mRNA and an increase in its degraded fragment were observed at the present of miR-263a mimic, indicating that the binding of miR-263a to the 5'UTR of *CathBL* could induce mRNA decay. This result is presented in new Fig. 2E, Lines 150-156 in the Results and Lines 631-644 in the Materials and Methods.

3. In Figure 3, it needs to be clarified whether the pre-miR263a or mature miR-263a are injected into plant cells. In addition, in Figure 3B, C, and D, the method for analyzing accumulation of miR263a should be provided in the figure legend, from northern blotting or deep sequencing. In addition, what's the means of the relative values in the Y axis?

Reponses: We only found the mature miR-263a are injected into plant cells. We specify this information in the paragraph (Lines 188-191) and the legend of Fig. 3. The method used in Figure 3B, C, and D is qPCR. The means of the relative values in the Y axis is the amount of miR-263a relative to that of OsU6 snRNA. We rewrite the legends of Figure 3B, C, and D as “qPCR analysis for the amount of miR-263a relative to that of OsU6 snRNA.....”.

4. In Figure 5, all the figure legend and result description are confusing and cannot support the conclusion that miR-263a elevates *GATA19* expression through intro cleavage. For instance, in 5E, there are some inconsistencies between the figure legend and labels in the figure. Is the result accumulation of *GATA19* or miR263a? It needs more evidence to show that the cleavage in the middle of intron could improve the mRNA accumulation.

Reponses: We are really sorry to have uploaded a wrong version of Fig. 5 and apologize for causing confusion of the reviewers. The correct version of Fig. 5 is supplied. In the new Fig. 5D, overexpression of miR-263a with a plasmid harboring *GATA19*'s first two exons and first intron (*Ex-In-Ex*) increased the exon transcripts in rice protoplasts. Mutations at the 17-nt binding site within the intron region (*Ex-InMT-Ex*) abolished the activation effect of miR-263a on *GATA19* exons. The new data further prove that the cleavage in the intron could improve the mRNA

accumulation. This result is complemented in Lines 266-267.

5. In Figure 4G and 5G, the virus disease symptoms should be shown.

Reponses: We took photos of disease symptoms from the two experiments and show the photos in the supplementary figure Fig. EV3F.

Minor Comments:

1: The disease incidence rates (only 0.4% or 0.8%) are too low, and it is something wrong with the unit of Y axis.

Reponses: Thank you for pointing out the mistake in the unit of Y axis. We correct this mistake in the new Fig. 4H.

2: In figure 6C, the negative control of GATA19-YN and empty YC is lack.

Reponses: We complement the negative control of GATA19-YN and empty YC in the new Fig. 6E as the reviewer suggested.

3: In the proposed model, it is better to label the 5' and 3' end of miR263a.

Reponses: We label the 5' and 3' end of miR-263a as the reviewer suggested.

Referee #2:

This manuscript presents a study about the function of one insect microRNA miR-263a on virus accumulation in insect vector and rice host. The miR-263a was considered to inhibits cathepsin B-mediated apoptosis to facilitate viral proliferation in midgut epithelial cells of insect. However, it has not been clearly explained why the expression of miR-263a in insect infected by virus significantly decreased.

Reponses: In our previous study (Zhao *et al.*, 2022, PLoS Pathogens), we clarified the inhibitory mechanisms of RSV to the expression of miR-263a. The NP of RSV binds the transcription factor YY1 to promote its inhibitory effect on miR-263a transcription probably due to enlargement of the steric hindrance to the binding of Pol II to the promoter with more YY1. At the same time, RSV-derived small RNA, vsR-

3397, directly targets the promoter region of miR-263a to downregulate miR-263a transcription. We briefly introduce this information in the Introduction, Lines 80-84.

Reference:

Zhao W, Li Q, Sun M, Xiao Y, Cui F. Interaction between endogenous microRNAs and virus-derived small RNAs controls viral replication in insect vectors. *PLoS Pathog.* 2022;18(7):e1010709.

In rice, this exogenous miR-263a might activate jasmonate signaling pathway by upregulates the expression of transcription factor GATA19 to interfere JAZ1 binding with MYC2, which leading to rice antiviral activity. However, it is essential for the authors to provide more experimental evidence.

Reponses: To clarify whether insect-injected miR-263a activates jasmonate signaling pathway, we measured the change of *GATA19* transcript levels and JA and SA contents in rice plants fed by nonviruliferous SBPHs with antagomir-263a or antagomir-NC injection. After 2 d of injection, SBPHs were allowed to feed on rice seedlings for 2 d. The rice seedlings were cultured for an additional day before collected for further measurement. We found that both miR-263a and *GATA19* transcript levels were significantly reduced in rice plants fed upon by SBPHs with antagomir-263a injection compared to those with antagomir-NC injection. Correspondingly, the JA content decreased significantly, whereas JA-Ile and SA contents did not vary. These results suggest that insect-injected miR-263a primarily influences the JA signaling pathway in rice under natural conditions. Therefore, we focus on the effect of miR-263a on the JA signaling pathway. These new results are presented in the new Fig. 6B and Fig. EV6, Results part (Lines 298-305) and Materials and Methods part (Lines 529-535). The description of SA is deleted in new Fig. 6A and Results part.

In addition, the Figures should be displayed in order. The order of Figure 2, 3 and 5 is somewhat disorderly, especially Figure 5, the marks in the paper do not correspond to the Figure, and Figure 5E is not mentioned in the paper.

Reponses: We are really sorry to have uploaded a wrong version of Fig. 5 and apologize for causing confusion of the reviewers. The correct version of Fig. 5 is supplied. The order of Fig. 2, 3 and 5 is rearranged for more clarity.

Authors found that miR-263a positively regulated the accumulation of viruses in insect vectors, but its expression decreased significantly in the intestines and salivary glands of viruliferous insects (Fig.1B). What was the expression level of miR-263a at different times when the virus infected insects? Is the expression of miR-263a inhibited after virus infection? In addition, the expression of miR-263a decreased significantly in the intestines and salivary glands of viruliferous insects, but there was no significant difference in the expression of mir-263a in the whole insects. Is the expression of miR-263a in other tissues high?

Reponses: The effect of RSV infection on miR-263a expression was clarified in our previous work (The following Fig A-C from Zhao et al., 2021, PLoS Pathogens). qPCR showed that no significant changes in miR-263a expression were detected in the whole body of viruliferous planthoppers compared to that in nonviruliferous planthoppers (Fig. A). However, a significant downregulation of miR-263a expression was observed in the gut and salivary glands of viruliferous insects, while the levels of miR-263a in brain, fat body, ovary, and testis remained unchanged (Fig. B). Furthermore, miR-263a expression was the highest in the gut compared to other tissues (Fig. B). After nonviruliferous planthoppers were infected with RSV for 2, 5, and 8 d, the expression of miR-263a in the gut was dramatically decreased concomitant to an increase in the virus load over time (Fig. C).

In the present study, we still found that the decline of the expression of miR-263a was insignificant in the whole bodies of 5 viruliferous insects using qPCR. We applied a northern blot assay to compare miR-263a levels in the whole bodies of 30 viruliferous and nonviruliferous SBPHs. The increment of sample sizes made the decline of miR-263a expression in the whole bodies of viruliferous SBPHs significant. This result is complemented in the new Fig. 1C, Results (Lines 107-110), and Materials and Methods (lines 648-649).

(from Zhao *et al.*, 2021, PLoS Pathogens)

Reference:

Zhao W, Yu J, Jiang F, Wang W, Kang L, Cui F. Coordination between terminal variation of the viral genome and insect microRNAs regulates rice stripe virus replication in insect vectors. *PLoS Pathog.* 2021;17(3):e1009424.

The detection of disease resistance of both miR-263a overexpression plants and GATA19 knockout plants showed significant differences in disease resistance after 12 days (Fig. 4H, Fig.5H), but miR-263a began to decline 6 days after it was transferred from insects to plants, and almost disappeared after 12 days (Fig. 3C). How to explain these results?

Reponses: In Fig 3C, the nonviruliferous SBPHs were allowed to feed on rice leaves 2 d and then removed from rice. The amount of miR-263a in rice kept constant within 6 d while nearly undetectable at 12 d without SBPH infestation. This experiment demonstrated the stability of miR-263a in rice once secreted in rice by SBPHs. In contrast, for the disease resistance experiments, we used OE263a transgenic strains

that continuously express miR-263a (Fig. 4H) and *GATA19* knockout plants (Fig. 5G), both of which were inoculated with RSV by viruliferous SBPHs for 7 d. The constant expression of miR-263a in the OE263a transgenic strains and the inhibition of *GATA19* expression in the knockout plants exhibit profound effects on RSV replication and disease incidence in rice. These two experiments do not conflict with each other. In the practical field conditions, large populations of viruliferous SBPHs feed on rice and miR-263a is secreted continuously and exists stably in rice, which resembles the situation of the OE263a transgenic rice.

miR-263a up-regulates the JA pathway, but down-regulates the SA pathway at the same time (Fig. 6C). If miR-263a negatively regulates the accumulation of viruses in rice from the SA pathway, these results need to be discussed.

Reponses: Based on the comment of Reviewer 1, we measured the change of *GATA19* transcript levels and JA and SA contents in rice plants fed by nonviruliferous SBPHs with antagomir-263a or antagomir-NC injection. After 2 d of injection, SBPHs were allowed to feed on rice seedlings for 2 d. The rice seedlings were cultured for an additional day before collected for further measurement. We found that both miR-263a and *GATA19* transcript levels were significantly reduced in rice plants fed upon by SBPHs with antagomir-263a injection compared to those with antagomir-NC injection. Correspondingly, the JA content decreased significantly, whereas JA-Ile and SA contents did not vary. These results suggest that insect-injected miR-263a primarily influences the JA signaling pathway in rice under natural conditions. Therefore, we focus on the effect of miR-263a on the JA signaling pathway. These new results are presented in the new Fig. 6B and Fig EV6, Results part (Lines 298-305) and Materials and Methods part (Lines 529-535). The description of SA is deleted in new Fig. 6A and Results part.

Major revision:

1. It can be seen from the Fig. 1A that rRNA of SBPH in line1 is degraded, this may have led to significant differences in pre-miR-263a and miR-263a between the two

insect groups. Interestingly, the expression of miR-263a decreased in the gut and salivary glands of viruliferous insects, while the decrease was not significant in the whole body. It is recommended to use northern to verify the expression level. The NP-RNA level of RSV increased significantly with the up-regulation of miR-263a level. This result indicated that miR-263a was conducive to the accumulation of RSV in SBPH, but there was no direct evidence to prove that it was involved in viral replication.

Reponses: In Fig. 1A, the two samples of SBPH or rice represent two replicates. We repeated this experiment with a better integrity of the rRNA and replaced the original figure.

In Fig. 1B, the decline of the expression of miR-263a was insignificant in the whole bodies of 5 viruliferous insects using qPCR. We applied a northern blot assay to compare miR-263a levels in the whole bodies of 30 viruliferous and nonviruliferous SBPHs. The increment of sample sizes made the decline of miR-263a expression in the whole bodies of viruliferous SBPHs significant. This result is complemented in the new Fig. 1C, Results (Lines 107-110), and Materials and Methods (lines 648-649).

Our previous work revealed that insect miR-263a promotes RSV replication in SBPHs through binding to the extended 3' terminal region of viral genomic RNA1 segment, thereby reducing the inhibitory effect of these extension sequences on viral promoter activity (Zhao *et al*, 2021). In this study, we found that miR-263a targets insect *CathBL* to inhibit apoptosis. We accept the reviewer's comment and replace "RSV replication" with "RSV accumulation" in the whole manuscript.

References:

Zhao W, Yu J, Jiang F, Wang W, Kang L, Cui F. Coordination between terminal variation of the viral genome and insect microRNAs regulates rice stripe virus replication in insect vectors. *PLoS Pathog*. 2021;17(3):e1009424.

2. Similarly in the results of line119 paragraph, there is no direct evidence that CathBL is involved in viral replication, which involves viral genome replication and

assembly, not just changes in NP level.

Reponses: We accept the reviewer's comment and replace "RSV replication" with "RSV accumulation" in the whole manuscript.

3. In Fig. 2G and 2J, the two figures need to be merged to see the cells clearly and make sure that the label is in the cell layer.

Reponses: The merged figures are added in the new Fig. 2H and 2K.

4. In Fig. 2H, GFP-treated insects exhibited predominantly nuclear localization of TUNEL staining, whereas DMSO treatment resulted in non-exclusive nuclear localization of TUNEL. Is TUNEL located in the nucleus? The presence of NP is barely discernible in the cellular layer following DMSO treatment, whereas it is predominantly localized within the intestinal epidermis. There is also no significant difference in NP between DMSO and CA-074 methyl ester treatment.

Reponses: In the early stage of apoptosis, the TUNEL signals are more prominent in the nuclei than in other cellular regions. The fluorescence distribution after DMSO treatment showed slight differences compared to the dsGFP group, possibly due to different apoptosis states within the cells. In midguts treated with DMSO, the signal of NP in the cytoplasm of midgut cells appears weak, representing the low viral load. In the CA-074 methyl ester-treated group, the NP signal in the cytoplasm is stronger, in accordance with the higher viral load and the inhibited apoptosis.

5. In Fig. 3E, the internal reference LsU6 snRNA and OsU6 snRNA of some samples were not hybridized, please explain the reason. In addition, there was the mobility shift of miR-263a in rice samples, which was not consistent with the result description. The two bar charts in Fig. 3F need to be distinguished.

Reponses: In Fig 3E, the 3rd, 4th, 8th, and 9th lanes from the left are the samples of *in vitro* synthesized miR-263a with a 3'-OH end (miR-263a-OH), not containing RNAs from rice leaves or SBPHs. Thus, no endogenous OsU6 or LsU6 is hybridized in these four lanes. The 1st and 2nd lanes are the rice samples fed on by SBPH, and no mobility

shift of miR-263a was observed after β -elimination treatment, indicating that miR-263a is methylated in rice. The two bar charts in Fig. 3F are labeled with “Nuclear” and “Cytoplasmic” as suggested by the reviewer.

6. How is the difference of NP relative to actin calculated in Fig. 4B? It is obvious from the Westernblot that the ratio is greater than 1, which is inconsistent with the bar figure. These data need to be checked and calculated.

Reponses: Thank you for pointing out this mistake in the Y-axis values. We correct this mistake in the new Fig. 4B.

7. There is a messy ordering problem that makes it difficult to read and understand in Fig. 5. The marks in the manuscript do not correspond to the figure. There are A-H in the figure and A-G in the manuscript. The Fig. 5E is not mentioned in the manuscript, and the 5E mentioned in the manuscript is actually 5F in the Figure.

Reponses: We are really sorry to have uploaded a wrong version of Fig. 5 and apologize for causing confusion of the reviewers. The correct version of Fig. 5 is supplied.

Referee #3:

1. Fig 2B; Two genes are not detected and cannot be concluded whether they are regulated or not with these panels? Or this has to be shown using a different scale on the X?

Reponses: The transcript levels of *Dsimw501* and *UGT2B5* are quite low, not at the same scale of other genes, and no significant difference was observed after injection of the agomir or antagomir of miR-263a compared to NC. As suggested by the reviewer, we use different scales on the Y-axis to show the transcript levels of the five genes in the new Fig. 2A and 2B.

2. Fig. 2B Why NTFYbL has a positive instead of a negative if this gene is a target?

Can you try to explain? OR if not discard this "target" too.

Reponses: miRNAs typically suppress the expression of target genes. However, there are instances where miRNAs upregulate the expression of target genes no matter binding the cds, 5' UTR, 3' UTR or promoter region (He *et al.*, 2016; Ørom *et al.*, 2008; Shi *et al.*, 2020; Place *et al.*, 2018). In Fig. 2B, *NTFYbL* seems to be positively regulated by miR-263a. We continue to verify the direct interaction of miR-263a with the target site of *NTFYbL* using dual luciferase assays in *Drosophila* S2 cells. This experiment excluded *NTFYbL* as possible target of miR-263a (Fig. EV2).

References:

He J, Chen Q, Wei Y, Jiang F, Yang M, Hao S, Guo X, Chen D, Kang L. MicroRNA-276 promotes egg-hatching synchrony by up-regulating *brm* in locusts. *Proc Natl Acad Sci U S A*. 2016;113(3):584-9.

Ørom UA, Nielsen FC, Lund AH. MicroRNA-10a binds the 5'UTR of ribosomal protein mRNAs and enhances their translation. *Mol Cell*. 2008;30(4):460-71.

Shi X, Liu TT, Yu XN, Balakrishnan A, Zhu HR, Guo HY, Zhang GC, Bilegsaikhan E, Sun JL, Song GQ, Weng SQ, Dong L, Ott M, Zhu JM, Shen XZ. microRNA-93-5p promotes hepatocellular carcinoma progression via a microRNA-93-5p/MAP3K2/c-Jun positive feedback circuit. *Oncogene*. 2020;39(35):5768-5781.

Place RF, Li LC, Pookot D, Noonan EJ, Dahiya R. MicroRNA-373 induces expression of genes with complementary promoter sequences. *Proc Natl Acad Sci U S A*. 2018;115(14):E3325.

3. Fig. 2C the differences are very minor at 10nm compare to Fig. 2D? Although I understand that the miR-resistant mutant experiment suggest that this difference is significant. This is the mutant version I previously mentioned that could be used to test the role of miR263 regulation in RSBV infection.

Reponses: We appreciate the understanding of the reviewer about the minor difference with the miR-263a mimic treatment in the dual luciferase assays and the significance of this difference reflected by the mutant. In our previous work, we also observed the modest impact of miRNAs on the target in the dual luciferase assays (Zhao *et al.*,

2022; Yu *et al.*, 2024).

References:

Zhao W, Li Q, Sun M, Xiao Y, Cui F. Interaction between endogenous microRNAs and virus-derived small RNAs controls viral replication in insect vectors. *PLoS Pathog.* 2022;18(7):e1010709.

Yu J, Zhao W, Chen X, Lu H, Xiao Y, Li Q, Luo L, Kang L, Cui F. A plant virus manipulates the long-winged morph of insect vectors. *Proc. Natl. Acad. Sci. U.S.A.* 2024; 121:e2315341121.

4. Fig. 3A is normalized by rRNA? There is large variation in U6 probes which are also constitutive. Are other miRNAs NOT transported to the host that can be used as controls for "transport".

Reponses: Fig. 3A demonstrates the existence of miR-263a in rice after SBPH infestation. We did not make normalization using rRNA or U6 because we did not quantify or compare the amount of miR-263a in rice and SBPH. In our lab, we use this experimental system to determine which SBPH miRNAs are transported to rice. miR-87 is found NOT transported to rice . Considering that this result has been published, we decide not to put it in the present manuscript.

5. Fig. 3B and 3C what do you mean by 0 levels in a qPCRs? There is no difference of miR263 levels between viliferous and non viliferous interactions? Any explanation?

Reponses: A “zero” indicates that no CT (Threshold Cycle) value is detected when quantifying the miR-263a expression in rice with qPCR. Although the expression of miR-263a is downregulated in the salivary glands of SBPH upon RSV infection, the

amounts of miR-263a secreted in rice is comparable between nonviruliferous and viruliferous SBPHs with continual infestation for 2, 4, or 6 d (Fig. 3D). We speculate that this might be due to the way miR-263a secreted out of salivary glands. Comparable amount of miR-263a could be loaded in saliva exosomes of viruliferous and nonviruliferous SBPHs for transportation.

6. Fig. 3D, this is the type of kinetics that needs to be done with plants expressing a miR-resistant version of the target to conclude about miR263 action on this target.

Reponses: Fig. 3D demonstrates that the amounts of miR-263a secreted in rice is comparable between nonviruliferous and viruliferous SBPHs with continual infestation for 2, 4, or 6 d.

This suggestion on expressing a miR-resistant version of the target to conclude about miR263 action on this target is constructive. We conducted this experiment in rice protoplasts. Overexpression of miR-263a with a plasmid harboring *GATA19*'s first two exons and first intron (*Ex-In-Ex*) increased the exon transcripts in rice protoplasts. Mutations at the 17-nt binding site within the intron region (*Ex-InMT-Ex*) (i.e., miR-263a resistant version of the target) abolished the activation effect of miR-263a on *GATA19* exons. The new data further prove that miR-263a acts on *GATA19*. This result is complemented in Lines 266-267 and as the new Fig 5D.

7. The experiments on protoplasts to show that *GATA19* is upregulated by miR263Oex in transient assays and plants support the idea that miR263 regulates *GATA19* and a version miR-resistant on the intron sequence could be used in the transient assay to test this regulation (even isoforms expression to see global impact on alternative splicing of this gene?). Clearly that a *GATA19* KO regulates defense genes (jasmonate pathway) can be through many mechanisms (it is a TF!) and a *GATA19* miResistant must reduce Jasmonate levels and clearly show that the targeting of miR263 for *GATA 19* is relevant *in vivo*.

Reponses: We agree with the reviewer. As mentioned above, we created miR-263a resistant version of the target. Mutations at the 17-nt binding site within the intron

region (*Ex-InMT-Ex*) abolished the activation effect of miR-263a on *GATA19* exons. The new data further prove that the cleavage in the middle of intron could improve the mRNA accumulation of *GATA19*. This result is complemented in Lines 266-267 and as the new Fig 5D.

Dear Dr. Cui,

Thank you for submitting a revised version of your manuscript. We have now received input from all original reviewers, who now find that their main concerns have been addressed satisfactorily. Therefore, I will be happy to accept the manuscript for publication after a textual revision in response to the remaining point by reviewer #1.

Additionally, there remain a few editorial points that need addressing before I can extend official acceptance of the manuscript:

1. Please submit a complete author checklist, which you can download from our author guidelines (<https://www.embopress.org/pb-assets/embo-site/EMBO%20Press%20Author%20Checklist-1642513524327.xlsx>). Please insert information in the checklist that is also reflected in the manuscript. The completed author checklist will also be part of the Review Process File.
2. Please rename "Conflict of interest" section into "Disclosure and competing interests statement" (further info: <https://www.embopress.org/page/journal/14602075/authorguide#conflictsofinterest>).
3. CRedit has replaced the traditional author contributions section because it offers a systematic, machine-readable author contributions format that allows for more effective research assessment. Please remove the Authors Contributions from the manuscript and use the free text boxes beneath each contributing author's name in our online submission system to add specific details on the author's contribution. More information is available in our guide to authors.
4. In the Data Availability section, please add a resolvable link to the PP716112 dataset. More information about the format of this section can be found here: <https://www.embopress.org/page/journal/14602075/authorguide#dataavailability>.
5. All Materials and Methods need to be described in the main text using our 'Structured Methods' format. According to this format, the Methods section includes a Reagents and Tools Table (listing key reagents, experimental models, software and relevant equipment and including their sources and relevant identifiers) followed by a Methods section describing the methods, ideally using a step-by-step protocol format. The aim is to facilitate adoption of the methodologies across labs. Please download and fill our Reagents and Tools Table template (.docx), which you can find in our author guidelines: <https://www.embopress.org/page/journal/14602075/authorguide#structuredmethods>. It appears that the information currently included in Appendix Table S1 could form a part of the Reagents and Tools table. When submitting your revised manuscript, please do not include the Reagents and Tools Table in the Methods section of the manuscript but upload it as a separate file choosing the file type "Reagent Table". An example of a Method paper with Structured Methods can be found here: <https://www.embopress.org/doi/10.15252/msb.20178071>.
6. Please remove EV table legends from the manuscript text file.
7. Please upload EV tables as Excel files rather than zip folders.
8. There is a callout to Table 3 in the manuscript text, please check and correct.
9. Our data editors have flagged the following issues in figure legends that need correcting:
 - Please define the annotated p values ****/**/*/* as well as provide the exact p-values for the same in the legend of figure EV2 as appropriate.
 - Please provide the exact p values in the legends of figures 1B-E; 2A, B, C, D, F, G, J; 3B, F; 4A, B, D, H; 5C, G; 6B; EV4, EV5 A, B; EV6.
 - Please indicate the statistical test used for data analysis in the legends of figures 4E, G; 5A, D, F; 6A.
 - Please provide information on the number and nature of replicates in the legends of figures 1B, D-E; 3F; 6A, B; EV2; EV3 A-E; EV4, EV5 A, B; EV6.
 - Please define the error bars in the legends of figures 2A, B, C, F, G, J; 3B, C, D, F; 4A, B, D, E, G, H; 6A, B; EV2, EV4, EV5 A, B; EV6.
10. Papers published in The EMBO Journal are accompanied online by a 'Synopsis' to enhance discoverability of the manuscript. It consists of A) a short (1-2 sentences) summary of the findings and their significance, B) 3-4 bullet points highlighting key results and C) a synopsis image that is 550x300-600 pixels large (width x height, jpeg or png format). You can either show a model or key data in the synopsis image. Please note that the image size is rather small and that text needs to be readable at the final size. Please send us this information together with the revised manuscript.

With best wishes,

leva

We realize that it is difficult to revise to a specific deadline. In the interest of protecting the conceptual advance provided by the work, we recommend a revision within 3 months (13th May 2025). Please discuss the revision progress ahead of this time with the editor if you require more time to complete the revisions. Use the link below to submit your revision:

Referee #1:

The authors reveal the dual roles of a insect miR-263a in virus infecting insects and plant cells. In insect cells, miR-263a enhances virus infection, whereas miR-263a inhibits virus infection in rice plants. Totally, the authors have satisfactorily answered the initially raised criticisms. However, some experimental method and conclusions are not well explained. Although the authors have explained the potential mechanism, it is still elusive why the miR-263a-targeted intron can increase the accumulation of mature GATA19 mRNA.

Referee #2:

Authors have revised the MS according to my comments, I recommend to accept it now.

Referee #3:

This is a revised version that has been improved and all my comments, notably the use of miR-resistant variants, have been included. The authors addressed all other comments in a satisfactory way.

All editorial and formatting issues were resolved by the authors.

Dear Feng,

Thank you for addressing the final editorial issues and clarifying the point about the source data. Thank you also for approving the final textual edits. I am now pleased to inform you that your manuscript has been accepted for publication in the EMBO Journal.

Please note that the link (URL) to the PP716112 dataset will have to be added in the Data Availability section latest at the proofs stage, even if it is not yet publicly accessible at that point. Otherwise, the typesetters will not be able to finish typesetting your article. If you have the dataset URL at hand, you can also forward it to me now, and I can include it in the manuscript text for you.

Finally, we would like to promote your manuscript among the Chinese readership. Therefore, we would like to invite you to prepare a short summary of the manuscript in Chinese (1500-2000 Chinese characters), which we will promote on the WeChat platform 'BioArt' with more than 610,000 followers.

If you are interested in this opportunity, we recommend covering the article very close to its online publication date. Thus, ideally we would very much appreciate if you could send us a draft within the next 7 working days. Please let us know whether or not you would be interested in contributing such a short summary in Chinese.

I have included below some general guidelines on how to prepare a summary and a link to recent examples for your reference. Please let me know if you have any questions about this.

If you have any questions, please do not hesitate to contact the Editorial Office. Thank you for this contribution to The EMBO Journal and congratulations on a nice study!

With best wishes,

Ieva

General WeChat Summary Guidelines

1. These summary articles are meant to be targeting general audience so please limit the use of specialized technical terms, acronyms and jargon.
2. A summary usually starts with brief background information of the reported work, which is followed by explaining the findings in some detail, and ends with a short review of the conclusions as well as the implications of the work and future directions for the research.
3. The summary should at least contain one graphical item, such as a scheme or a figure from the paper.
4. Please provide ONE SINGLE document containing all text and graphical materials, ideally as a Word.docx or .doc file. Please

DO NOT provide the document as a .pdf file.

5. Please DO NOT publicly release the document before the paper is officially published online.

Summary Examples

EMBO J | 罗招庆/欧阳松应揭示谷酰胺脱氨酶MvcA的去泛素化功能

EMBO J | 王松灵院士团队揭示组织内应力调控大型哺乳动物乳恒牙替换的新机制
